# Conformational restriction shapes the inhibition of a multidrug efflux adaptor protein

Benjamin Russell Lewis [1], Muhammad R. Uddin[2], Mohammad Moniruzzaman [2], Katie M. Kuo[3], Anna J. Higgins[4], Laila M. N. Shah [1], Frank Sobott [4], Jerry M. Parks [5], Dietmar Hammerschmid [1], James C. Gumbart [3,6] ✉, Helen I. Zgurskaya [2] ✉ & Eamonn Reading [1] ✉

Membrane efflux pumps play a major role in bacterial multidrug resistance. The tripartite multidrug efflux pump system from *Escherichia coli*, AcrAB-TolC, is a target for inhibition to lessen resistance development and restore antibiotic efficacy, with homologs in other ESKAPE pathogens. Here, we rationalize a mechanism of inhibition against the periplasmic adaptor protein, AcrA, using a combination of hydrogen/deuterium exchange mass spectrometry, cellular efflux assays, and molecular dynamics simulations. We define the structural dynamics of AcrA and find that an inhibitor can inflict long-range stabilisation across all four of its domains, whereas an interacting efflux substrate has minimal effect. Our results support a model where an inhibitor forms a molecular wedge within a cleft between the lipoyl and αβ barrel domains of AcrA, diminishing its conformational transmission of drug-evoked signals from AcrB to TolC. This work provides molecular insights into multidrug adaptor protein function which could be valuable for developing antimicrobial therapeutics.

Multidrug resistance refers to the ability of bacterial pathogens to survive lethal doses from many structurally diverse compounds[1]. Bacterial multidrug resistance continues to spread at alarming rates, threatening human health globally. In 2019, bacterial multidrug resistance directly caused 1.27 million deaths worldwide, which was more than HIV and malaria combined[2].

A major mechanism of multidrug resistance is the activity of efflux pumps[3,4]. Efflux pumps are commonly overexpressed in response to antibiotic exposure, and can export a wide range of chemically diverse compounds, lowering intracellular antibiotic concentration and conferring drug resistance[1]. The Hydrophobic and Amphiphilic Resistance Nodulation and Cell Division (HAE-

RND) family of transporters play a key role in bacterial multidrug resistance. *Escherichia coli* AcrAB-TolC is the prototypical member of this family with homologues across other gram-negative ESKAPE bacteria[5–7]. It is a tripartite protein complex which spans the membrane envelope of gram-negative bacteria, where AcrB is the inner membrane transporter of the complex, AcrA the periplasmic adaptor protein (PAP) from the Membrane Fusion Protein (MFP) family of proteins, and TolC the outer membrane channel (Fig. 1a). Energised by the proton motive force, AcrB transports substrates, including antibiotics, from the intracellular environment to the outside of the cell, through a sealed channel formed by AcrA and TolC (Fig. 1a)[8,9].

[1]Department of Chemistry, King's College London, Britannia House, 7 Trinity Street, London SE1 1DB, UK. [2]Department of Chemistry and Biochemistry, University of Oklahoma, 101 Stephenson Parkway, Norman, OK 73019, USA. [3]School of Chemistry and Biochemistry, Georgia Institute of Technology, 837 State Street NW, Atlanta, GA 30332, USA. [4]School of Molecular and Cellular Biology & Astbury Centre for Structural Molecular Biology, University of Leeds, Leeds, UK. [5]Bioscience Division, Oak Ridge National Laboratory, 1 Bethel Valley Road, Oak Ridge, TN 37831, USA. [6]School of Physics, Georgia Institute of Technology, 837 State Street NW, Atlanta, GA 30332, USA. ✉e-mail: gumbart@physics.gatech.edu; elenaz@ou.edu; eamonn.reading@kcl.ac.uk

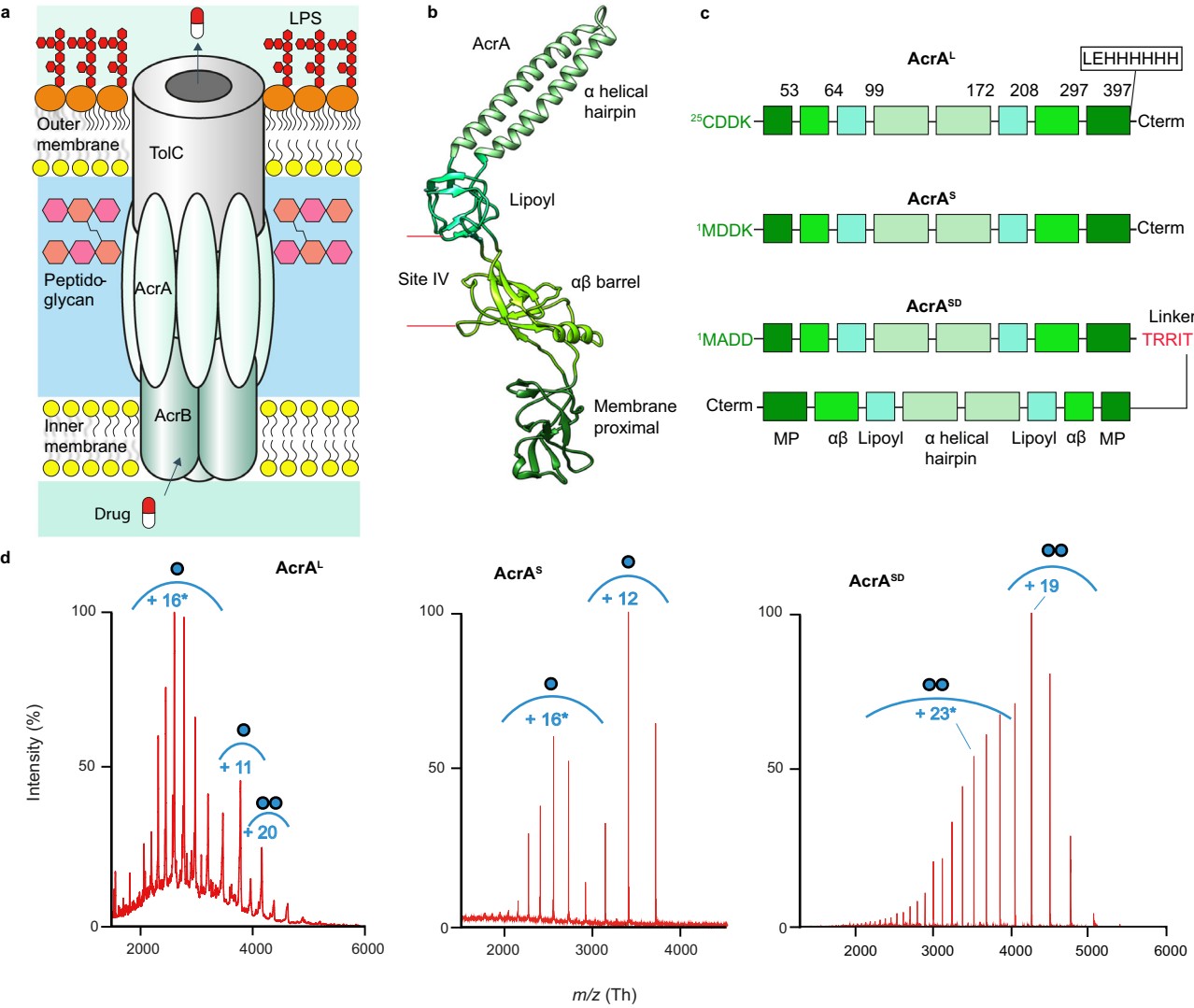

**Fig. 1 | AcrA organisation, structure, and constructs. a** Schematic of AcrAB-TolC complex embedded in the cell envelope. LPS Lipopolysaccharide. **b** Structure of AcrA isolated from the entire AcrAB-TolC complex (PDB:5O66)[23]. **c** Three constructs of AcrA. AcrA[L] contains Cys25 which is lipidated after signal peptide 1-24 cleavage. AcrA[S] contains a Cys25Met mutation, and therefore lacks the lipidation and signal peptide. AcrA[SD] contains two AcrA[S] sequences connected by a TRRIT linker. **d** Native-MS characterisation of the AcrA constructs at pH 6.0. Proteins buffer was exchanged in 100 mM ammonium acetate buffer prior to MS. AcrA[L]

required the presence of 2× critical micelle concentration (CMC) of DDM at 0.03%. AcrA[L] presents as a mix of monomers and dimers, AcrA[S] as monomers and AcrA[SD] as a dimer. Spectra collected on biological replicates and by two different native-MS systems, a time of flight and an orbitrap system, provide confidence these spectral signatures are not artefacts of protein preparation or MS detection[67]. Envelopes marked with an asterisk (*) represent the higher charge state indicative of intrinsic disorder. Masses found in Supplementary Table 1.

Efforts have been made to develop efflux pump inhibitors (EPIs) against these systems to 'revive' the activities of various pre-existing antibiotics, which a bacterial population has become resistant to[10]. Until now, the focus has been to generate EPIs targeting the AcrB transporter itself. However, previously identified EPIs have failed to progress to clinical trials, due to toxicity issues and the promiscuous nature of AcrB to transport its inhibitors[11,12]. Thus, there is a need to explore other avenues in the quest to generate successful EPIs, with AcrA emerging as a potential target for inhibition[10,13]. Recently, NSC 60339 was identified as an AcrA inhibitor through a joint experimental-computational screen[10]. Previous work suggested NSC 60339 could cause a conformational change in AcrA, and proposed a possible binding site at the interface between the lipoyl and αβ barrel domains, termed site IV (Fig. 1b)[10,14]. Moreover, AcrA has recently been shown to have more diverse functions, being identified as a bacterial 'necrosignal' within *E. coli* swarms[15]: when a subpopulation of the swarm dies, dead cells release the necrosignal AcrA, which binds TolC on the

outside of other live cells, stimulating efflux within the affected area and the upregulation of various efflux pumps[16]. Interestingly, recent work has also shown that NSC 60339 can inhibit AcrA-mediated necrosignalling ability[16]. Understanding the molecular mechanisms of AcrA inhibition is therefore important for future EPI and necrosignalling drug discovery. However, a mechanism of AcrA inhibition remains elusive.

AcrA acts as a flexible adaptor protein between AcrB and TolC, with its elongated shape essential for this function[17–19]. It has been proposed to form a trimer of dimers in the assembled complex to maintain a sealed channel, and accommodate the conformational movement of AcrB as it cycles through its rotational efflux mechanism[20]. After signal peptide cleavage AcrA is lipidated at Cys25, which anchors it to the inner membrane, although previous studies have shown that the presence of the lipid moiety is not required for AcrA function[17,18,21,22]. AcrA consists of four linearly arranged domains, the α-helical hairpin, lipoyl, αβ barrel and membrane proximal (MP)

domains, each connected by small flexible linkers (Fig. 1b). The αβ barrel and MP domains interact with AcrB, whereas the α-helical domain interacts with the periplasmic, α-helical coiled coils of TolC, where it is proposed to regulate its 'open' and 'closed' states[23].

Here, we used a combination of native and hydrogen/deuterium exchange mass spectrometry (nMS and HDX-MS), molecular dynamics (MD) simulations, biophysical and cellular efflux assays to determine a mechanism of action for the AcrA inhibitor, NSC 60339[10,14]. Our results suggest that NSC 60339 acts as a molecular wedge within a cleft between the lipoyl and αβ barrel domains of AcrA, reducing its structural dynamics across all four domains. Furthermore, we investigated an efflux substrate, novobiocin, which is known to bind to AcrA, but does not inhibit efflux, revealing minimal change in AcrA dynamics in the presence of novobiocin. We also performed in vivo efflux assays with AcrA mutants, to investigate whether deliberately targeting AcrA at the lipoyl-αβ barrel and the αβ barrel-MP flexible linkers could affect efflux. This confirmed the lipoyl-αβ cleft as a druggable site for targeting AcrA during efflux and revealed a possible site between the αβ barrel-MP domains for future drug design. Overall, our results suggest that NSC 60339 inhibits efflux via restriction of AcrA structural dynamics, which could reduce the efficiency of AcrAB-TolC functional rotation and possibly perturb interactions within the complex itself during efflux and necrosignalling.

## Results

To best mimic the periplasmic environment of AcrA - the cytosol is on average 1.7 pH units higher than the external environment—we performed our investigations at pH 6.0[24–26]. This will also ensure that our findings are consistent with previous microbiological and biochemical work performed on NSC 60339, which was completed at pH 6.0[10,14,27]. Consequently, our HDX labelling times were amended to accurately represent typical labelling times at physiological pH (see Methods), as pH influences the exchange rates of HDX.

### AcrA consists of both structured and disordered regions

To understand AcrA structural dynamics we optimised our sample for investigation by HDX-MS, which provides molecular level information on protein conformation and dynamics[28]. HDX occurs when amide hydrogens on the peptide backbone become accessible to deuterated buffer, and undergo isotopic exchange with $D_2O$. Accessibility results from structural unfolding and hydrogen bond breakage, thus unstructured regions exhibit rapid exchange when exposed to solvent, whereas regions with stable secondary structures exhibit slower exchange as these regions are protected and stabilised by hydrogen bonding networks. To enable reliable interpretation of the effect of inhibitor and substrate interactions on AcrA dynamics, we explored constructs that would ensure sample homogeneity (Fig. 1c). We monitored oligomeric homogeneity by nMS, which allows intact proteins to be transferred to the gas phase while preserving non-covalent interactions and, thus, is adept at reporting on protein oligomeric states[29,30].

We first purified AcrA containing its signal peptide (residues 1–24), which is cleaved after lipidation at Cys25 (AcrA[L]) (Fig. 1c)[27]. This construct is associated with the membrane and was therefore purified from the membrane in $n$-dodecyl-β-$D$-maltopyranoside (DDM) detergent micelles. nMS revealed AcrA[L] existed as a mixture of monomers and dimers at pH 6.0 (Fig. 1d); at pH 7.4 AcrA[L] had increased heterogeneity, forming monomers and up to pentamers (Supplementary Fig. 1). This degree of heterogeneity would complicate HDX-MS interpretation, therefore, we considered AcrA lacking the lipidation (AcrA[S]), which has previously been shown to retain its functionality and has been utilised in several other studies[16–18,27]. AcrA[S] lacks its signal peptide (residues 1–24) and has its Cys25 replaced by a starting methionine (Met25). It is expressed and purified from the cytosol and lacks lipidation on its N-terminus. nMS studies of AcrA[S] revealed it was

entirely monomeric at both pH 7.4 and 6.0 (Fig. 1d and Supplementary Fig. 1). Therefore, we decided to proceed with the AcrA[S] construct for our investigations at pH 6.0.

As AcrA forms a trimer of dimers in the assembled complex[8], we also explored an AcrA pseudo-dimer construct (AcrA[SD]) (Fig. 1c). This will enable us to probe the effect of dimerisation on AcrA structural dynamics, and how NSC 60339 inhibition is affected by the presence of a dimer interface. The construct contained two AcrA[S] sequences connected by a five amino acid linker, and nMS confirms it to be a homogenous pseudo-dimer at pH 6.0 (Fig. 1d).

Aside from elucidating stoichiometry, charge state distributions (CSDs) seen in nMS spectra can report on conformational properties[31,32]. Interestingly, our nMS spectra indicate AcrA could contain areas of intrinsic disorder in all constructs. nMS spectra of all three constructs present at least two CSDs, reflecting populations of folded and more unstructured conformers occurring during protein ionisation into the gas phase (Fig. 1d and Supplementary Fig. 1). It has been found that mixed CSDs such as these are diagnostic of folded proteins containing areas of intrinsic disorder[33]. This area is likely to be the MP domain as previously postulated, as it is often unresolved in crystal structures[18,34].

Next, we used HDX-MS to assess the intrinsic dynamics of our AcrA constructs. After quench and digestion optimisation on AcrA[S] (see 'Methods' and Supplementary Fig. 2), we could achieve > 95% sequence coverage for our HDX-MS investigations. A relative fractional deuterium uptake analysis of AcrA[S] reveals areas with time-dependent exchange, characteristic of a folded protein with differences in secondary structure and dynamics (Supplementary Fig. 3). The α-helices show a strong level of protection throughout the entire HDX time course, suggesting this is the most structurally stable area of AcrA. However, peptides within areas of the MP (residues 25–42, 375–380) and αβ barrel domains (residue 264-275) demonstrated near-maximum deuterium incorporation even at the earliest time points (10 s), which is indicative of unstructured regions[35]. This further supports that the MP domain predominantly contains areas of intrinsic disorder. Expanding on this notion further, we evaluated the structure of AcrA as predicted by AlphaFold2 (Supplementary Fig. 4)[36,37]. AlphaFold2 provides a per-residue confidence score (pLDDT) between 0-100 for each residue; regions with a score of < 50 may be unstructured. Regions 1–36 and 379–397 are both in the MP domain and contain many residues with a pLDDT score < 50. This is in agreement with our mass spectrometry results that the MP domain contains unstructured regions. Thus, we classify AcrA as a folded protein with areas of intrinsic disorder. This likely benefits AcrA functionally as a PAP, as the periplasm is a dynamic environment that can change size under different conditions, and AcrA is required to be flexible enough to accommodate these changes to maintain a sealed channel in the assembled complex[38].

### An efflux pump inhibitor, NSC 60339, causes long-range restriction of the AcrA backbone

To explore the effects of NSC 60339 inhibition on AcrA[S], we performed differential HDX (ΔHDX) between AcrA[S] + NSC 60339 and AcrA[S] alone. ΔHDX analyses the difference between two states and is a sensitive approach to characterise, and localise, the effect of a condition on the structural dynamics of a protein[39]. For a given peptide, we calculate the amount of deuterium incorporated in each protein state (no drug/NSC 60339/novobiocin) by comparing the centroid $m/z$ for a given time point to a non-deuterated reference sample. In Fig. 2a, we can see that for a representative peptide (residues 308–323), after 1 min the amount of deuterium incorporated is the same for AcrA[S] without drug and with novobiocin, yet in the presence of NSC 60339 there is > 1 Da reduction in deuterium uptake.

In the presence of NSC 60339, AcrA[S] exhibited stabilisation across all four domains (Fig. 2b, c), with the site IV region—identified as a

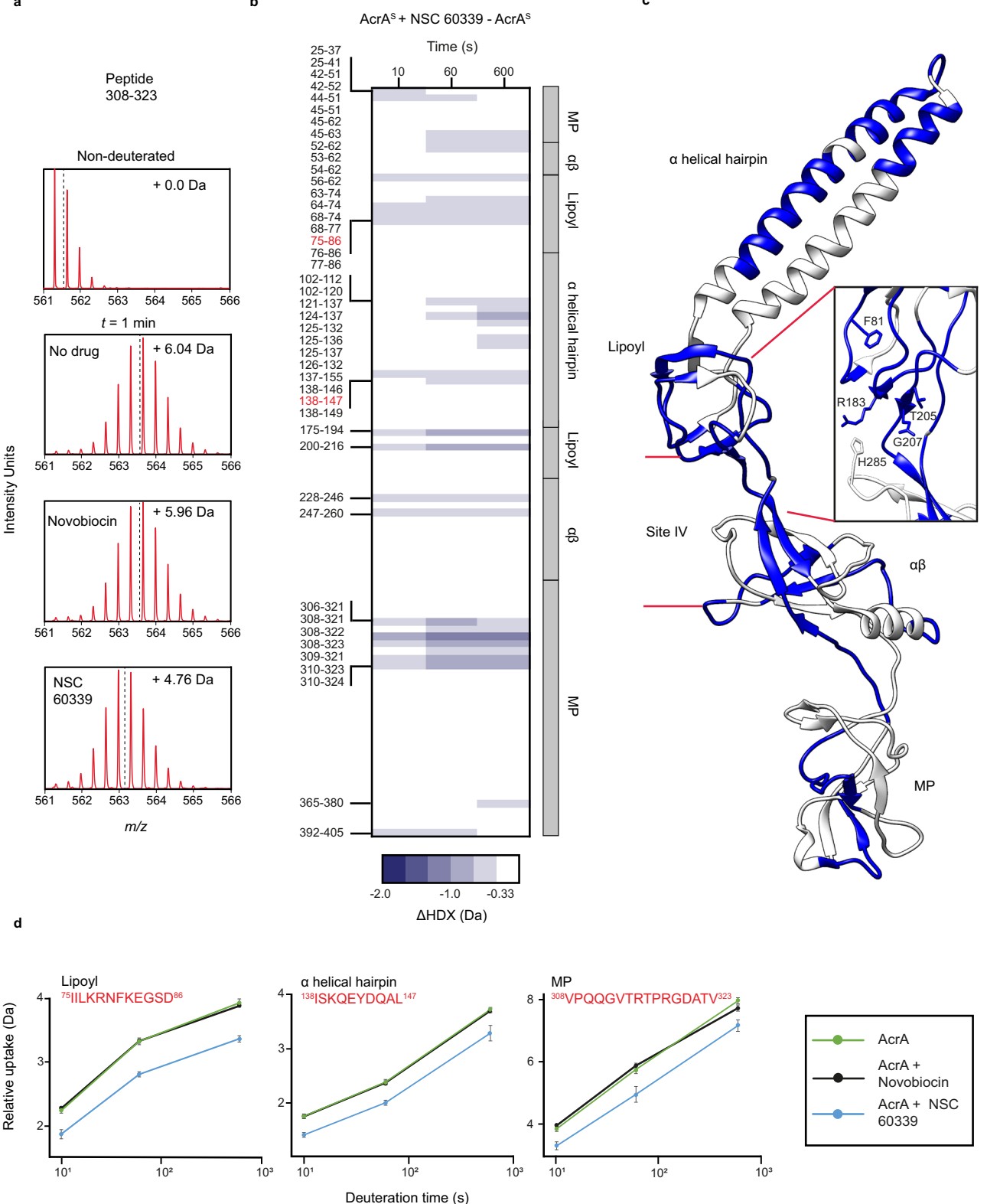

**Fig. 2 | The effect of NSC 60339 on AcrA$^S$ structural dynamics. a** The $m/z$ spectrum for peptide 308-323 under non-deuterating conditions and deuterating conditions with DMSO, NSC 60339 and novobiocin. The centroid is represented by the dotted line, and the mass change of the deuterated samples is written in Daltons. **b** Chiclet plot displaying the differential HDX (ΔHDX) plots for AcrA$^S$ +/− NSC 60339 for all time points collected. Blue signifies areas with decreased HDX between states. We defined significance to be ± ≥ 0.33 Da change (see 'Methods' and Supplementary Fig. 9) with a $P$ value ≤ 0.01 in a two-sided Welch's $t$ test

(8 independent measurements: $n_{biological} = 2$ and $n_{technical} = 4$). White areas represent regions with insignificant ΔHDX. Source data are provided as a Source data file. **c** ΔHDX for ((AcrA$^S$ + NSC 60339) − AcrA$^S$) for the latest time point is painted onto the AcrA structure (PDB:5O66) using HDeXplosion and Chimera[23,55,56]. Zoomed in insert of site IV is shown, with the side chains of implicated residues highlighted[14]. **d** Uptake plots for three peptides in different domains of AcrA. Uptake plots are the average deuterium uptake and error bars indicate the standard deviation (8 independent measurements: $n_{biological} = 2$ and $n_{technical} = 4$).

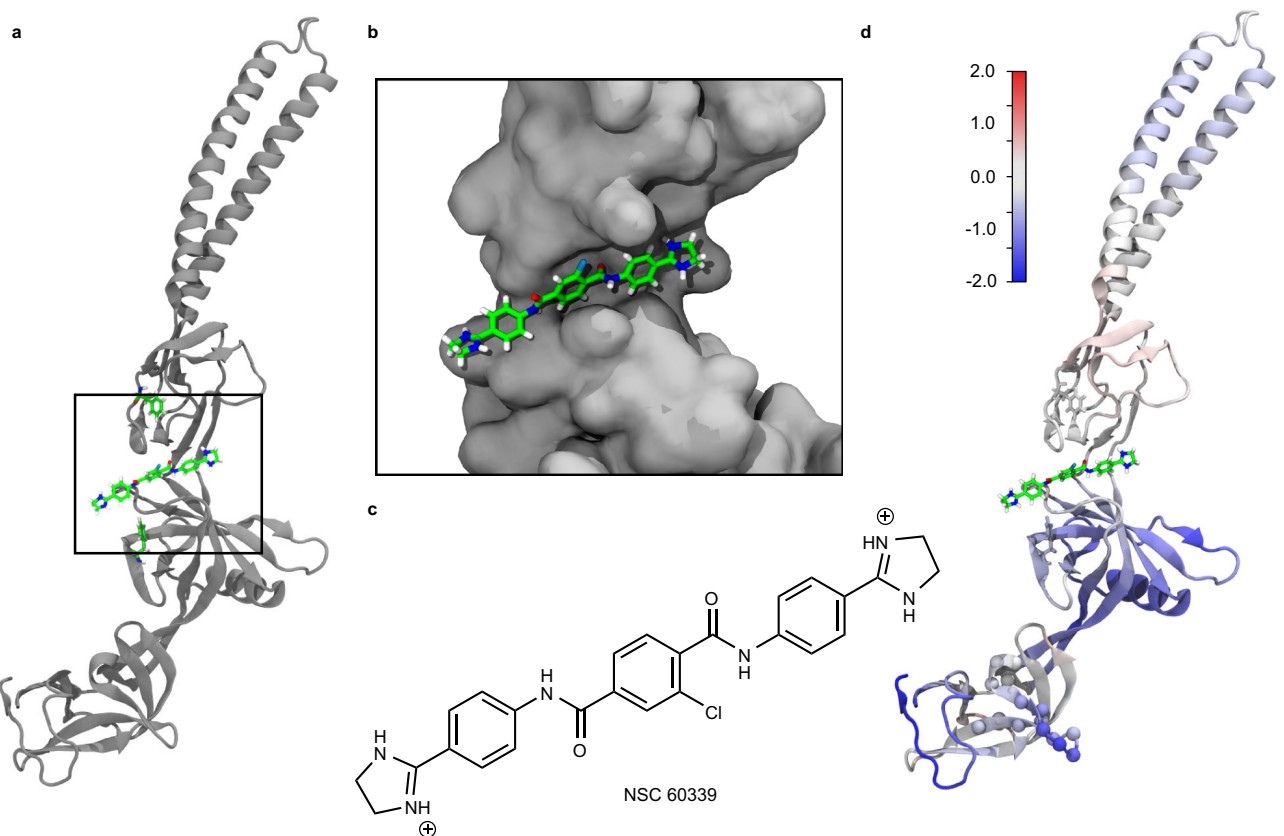

**Fig. 3 | MD simulations of NSC 60339 bound to AcrA$^S$. a** Docked structure of NSC 60339 (centre) to AcrA from Darzynkiewicz et al.[14]. F81 and F254 are shown above and below NSC 60339, respectively. **b** Zoomed-in view of the NSC 60339 binding pocket cleft. AcrA is shown in a surface representation. **c** NSC 60339 structure; calculated p$Ka$ values for the dihydroimidazoline groups are > 9, suggesting that it is dicationic at pH 6.0[68]. **d** AcrA coloured according to the difference in root-mean-square fluctuations (RMSF) between simulations of the bound and apo states, averaged over four replicas for each. Red indicates that the RMSF is greater in the bound state while blue indicates it's greater in the apo state (the colour range is from −2 Å, blue, to 2 Å, red as indicated by the colour bar). RMSF was calculated over the last 70 ns of each 100-ns simulation.

possible binding site within a cleft between lipoyl and αβ barrel domains by computational modelling and Trp fluorescence spectroscopy[14]–being affected (Figs. 2c and 3a, b). Portions of site IV exhibited a significant decrease in deuterium uptake in the presence of NSC 60339 (Fig. 2b–d), with peptides including residues F81 and F254 showing decreased uptake. Residues F81 (lipoyl domain) and F254 (αβ barrel domain) are both implicated in NSC 60339 binding with F81 also being essential for efflux function. HDX reduction within these areas may be due to drug binding stabilising the local area through direct interactions with amide hydrogens, or a reduction in solvent accessibility within the implicated domains. Based on the docked structure (Fig. 3a), F81 and F254 bracket site IV but do not make direct contact with NSC 60339. However, in MD simulations, F254 shows reduced fluctuations (Fig. 3d) and both F81 and F254 show reduced solvent-accessible surface area (SASA; Supplementary Fig. 5), demonstrating how binding of NSC 60339 can have allosteric effects on different regions of AcrA. Our HDX-MS and MD data, together with the previous work, supports that site IV is the binding site of NSC 60339.

Aside from site IV, our HDX results reveal that NSC 60339 induces long range stabilisation of AcrA structural dynamics away from the NSC 60339 binding site. The α-helical hairpins show stabilisation on both helices and the flexible linker, primarily in the later time points. This is likely due to the stabilisation exhibited by the lipoyl domain due to NSC 60339, restricting the ability of AcrA to freely position the helices, possibly reducing its conformational plasticity. Additionally, the MP domain exhibits extensive protection in the presence of NSC 60339. The largest degree of protection in the MP domain occurs between residues 308–324, with multiple peptides within this region

showing a sustained protection throughout the entire HDX time course (Fig. 2b, c). Our findings support that the effects seen in these domains are a downstream effect of NSC 60339 rather than secondary binding sites. Root Mean Square Fluctuations (RMSF) from our MD simulations of the bound and apo states show a high degree of stabilisation of residues 308–324 even compared to other residues (Fig. 3d and Supplementary Fig. 6), supporting our HDX data and providing confidence that NSC 60339 can cause long range stabilisation of dynamics away from the proposed binding site. It has also been shown that AcrA lacking the MP domain is still bound by NSC 60339, advocating that its stabilisation is allosteric to NSC 60339 binding to site IV[14]. In contrast to the MP domain, the difference in RMSF for the lipoyl domain was modest (Fig. 3d and Supplementary Fig. 6); however, portions of the lipoyl domain did see reduced SASA, possibly explaining the increased protection observed by HDX-MS (Supplementary Fig. 5). Interestingly, some areas of the MP, lipoyl and αβ barrel domains experienced increased SASA in the MD simulations but showed reduced RMSF or HDX (Supplementary Fig. 5). This observation supports that, even with increased solvent accessibility, enhanced protection of the backbone amide to HDX in these regions is likely dominated by its restricted structural dynamics.

As a control and a comparison to NSC 60339, we investigated the effect of novobiocin on AcrA$^S$ dynamics. Novobiocin is a substrate of AcrAB-TolC, binds to AcrA and yet does not classify as an EPI[10]. As before, we performed differential HDX on AcrA$^S$ + novobiocin and AcrA$^S$ alone (Fig. 2d and Supplementary Fig. 7). Our results show that novobiocin has little observed effect on AcrA structural dynamics within our HDX experiments, which is in stark contrast to NSC 60339.

The peptide uptake plots in Fig. 2d show three peptides across three domains that exhibit statistically significant protection when inhibited by NSC 60339 (blue plot). The black plot represents the uptake for a peptide in the presence of novobiocin, and the green plot represents AcrA$^S$ alone. As the uptake plots show, there is no difference in deuterium incorporation between AcrA$^S$ alone and with novobiocin, even though the percentage of protein:ligand complex was the same (see Methods). As a supporting experiment, we monitored novobiocin binding to AcrA$^S$ using nMS (Supplementary Fig. 8 and Supplementary Table 2) and observed the protein:ligand complex. Overall, we observe that while it is a substrate of AcrA, novobiocin does not have global effects on structural dynamics like NSC 60339 (Supplementary Fig. 9). This supports our premise that NSC 60339 is a successful inhibitor of AcrA due to its ability to stabilise AcrA across all four of its domains. AcrA relies on its dynamism and flexibility to function as a PAP[19], so restricting this conformational ability may be a promising target for inhibition in general.

## NSC 60339 acts in a similar fashion against an AcrA pseudo-dimer

To characterise the effect of dimerisation on AcrA structural dynamics we performed differential HDX between the monomer and dimer (Supplementary Fig. 10). AcrA$^{SD}$ experienced areas of decreased HDX compared to AcrA$^S$, and therefore a stabilisation across all four domains. The area that exhibited the largest decrease in ΔHDX was the α-helical hairpins in the latest time point (10 min). This protection could be explained by complementary packing of the α-helical hairpins in the dimer leading to a stabilisation of its structure and possible solvent exclusion in the binding interface[40]. The MP domain also exhibited protection at the later time points between residues 306–342 and the αβ barrel (residues 261–288) in the early time points (10–60 s), suggesting dimerisation may also provide more structural order to these regions.

Our HDX-MS investigation shows that the AcrA$^{SD}$ has unique dynamics compared to the monomer, even though it possesses similar thermal stability to the AcrA$^S$ construct, as judged by circular dichroism (CD, Supplementary Fig. 11). This may aid in its association in a complex with AcrB. To test this, we compared the binding of AcrA$^S$ and AcrA$^{SD}$ to AcrB in SMALP (Styrene Maleic Acid Lipid Particles) nanodiscs using SMA-PAGE, which is a native-PAGE method adapted for use with SMALPs (Supplementary Fig. 12)[41,42]. It yields stoichiometric information and can reveal protein complex formation. AcrA$^S$ appears to bind trimeric AcrB SMALPs in a 1:1 model, as previously reported, yet AcrA$^{SD}$ appears to bind trimeric AcrB in a range of different stoichiometries, which were not resolved by SMA-PAGE analysis, suggesting dimerisation promotes higher-order AcrA interaction(s) with trimeric AcrB embedded in a lipid environment[27].

To characterise AcrA$^{SD}$ interactions with NSC 60339, we used a surface plasmon resonance (SPR) binding assay. For this purpose, AcrA$^S$ and AcrA$^{SD}$ were immobilised onto a chip at similar densities of 4743 and 4054 response units (RU), respectively and increasing concentrations of NSC 60339 from 6 μM to 200 μM were injected over the immobilised proteins. This was an effective way to qualitatively confirm similar binding of NSC 60339 to the two AcrA variants which suggests that pseudo-dimerisation does not affect the mode of interaction of NSC 60339 with the AcrA binding site (Supplementary Fig. 11).

To investigate if AcrA dimerisation has an effect on NSC 60339 inhibition, we repeated our differential HDX experiments with the AcrA$^{SD}$ construct (Fig. 1c). Our results show that NSC 60339 acts in a similar way when AcrA is a dimer (Fig. 4a). There is statistically significant protection across all four domains as seen with AcrA$^S$, with pronounced HDX protection across the lipoyl domain, where site IV is located, suggesting the drug is acting in a similar area. Furthermore, AcrA$^{SD}$ exhibits extensive protection in the MP domain, with the core of

this stabilisation happening at the same region (residues 306–324). There is also a stabilisation across the α2 helix in the α-helical hairpin; however, fewer peptides and a smaller region compared to AcrA$^S$. One possibility for this may be due to the packing of the hairpins in the dimer already making the region more stable compared to the monomer (Supplementary Fig. 10). Thus, the hairpins may be less effected by NSC 60339 inhibition in the dimer.

Interestingly, we see there is a reduced region of the αβ barrel domain being stabilised, on the other side of site IV. When AcrA$^{SD}$ is inhibited with NSC 60339, only residues 232–246 and 249–261 show significant protection, whereas AcrA$^S$ shows protection between residues 228–260 and 52–62, which may suggest a difference in how the drug is interacting within the binding site.

Overall, NSC 60339 appears to be inhibiting AcrA$^{SD}$ in a similar way, by reducing AcrA's structural dynamics across the entire protein. To support this hypothesis, we also ran simulations of the bound and apo states of a modelled AcrA$^{SD}$ starting from an existing crystal structure (PDB 2F1M)[18]. A notable reduction in the RMSF for the bound state was observed over most of both copies of the protein compared to the apo state (Fig. 4c and Supplementary Fig. 13).

As before, we also investigated the effect of novobiocin on AcrA$^{SD}$. Novobiocin appeared to have little effect on the structural dynamics of AcrA$^{SD}$, consistent with what our results showed for AcrA$^S$ (Supplementary Fig. 14). As with AcrA$^S$, we were able to capture the protein:ligand complex during nMS (Supplementary Fig. 8 and Supplementary Table 2).

## Specific targeting of AcrA flexible domain linkers impairs efflux ability in vivo

To determine whether targeting the flexible hinge regions of AcrA could lead to inhibition of antibiotic efflux by AcrAB-TolC in vivo, we next developed a covalent efflux inhibition assay using live *E. coli* cells. For this purpose, we constructed the plasmid borne *acrAB* with *acrA* variants containing unique Cys substitutions in the flexible linkers between the MP and αβ barrel (Leu50Cys, Ile52Cys, Arg225 Cys, Glu229Cys, Asn232Cys) and between the αβ barrel and lipoyl domains (Arg183Cys, Thr205Cys, Asp284Cys) (Fig. 5c). Cells producing the WT and AcrA(Cys)B variants were treated with a Cys-reactive probe MTS-rhodamine 6G (MTS-R6G). This probe is a substrate of AcrAB-TolC and its covalent binding to Cys residues of AcrA located in positions important for conformational flexibility and/or function of the protein is expected to inhibit efflux activity of AcrAB-TolC pump. In this assay, we used hyperporinated *E. coli* Δ9-Pore cells lacking all nine TolC-dependent transporters and carrying the plasmid borne AcrA(Cys) AcrB variants[43]. Hyperporination of the OM has eliminated the permeability barrier of the OM and facilitated MTS penetration into the periplasm to reach AcrA(Cys). Cells were pre-treated with MTS, the unreacted probe washed away and the kinetics of efflux of a fluorescent substrate Hoechst 33342 (Hoechst) was analysed.

We found that the MTS-R6G pre-treatment of cells producing wildtype AcrAB-TolC or carrying an empty vector did not affect the Hoechst accumulation, suggesting that the two intrinsic Cys residues of AcrB located in the transmembrane domain of the protein do not affect the outcome of the assay (Fig. 5b). We next compared Hoechst accumulation levels in the cells producing different AcrA(Cys) variants and with and without pre-treatment with MTS-R6G. All pumps assembled with AcrA(Cys) variants were as efficient in efflux of Hoechst as the wildtype pump, which is consistent with their ability to protect Δ9-Pore cells from novobiocin, erythromycin and SDS (Supplementary Table 3). The exception is AcrA(Leu50Cys) which appeared to be fully functional in MIC measurement (Supplementary Table 3) but was only partially efficient in efflux of Hoechst (Fig. 5b).

The pre-treatment with MTS-R6G inhibited the activity of AcrA Leu50Cys, Thr205Cys and Asn232Cys variants only, whereas no significant inhibition was seen for other analysed AcrA(Cys) variants

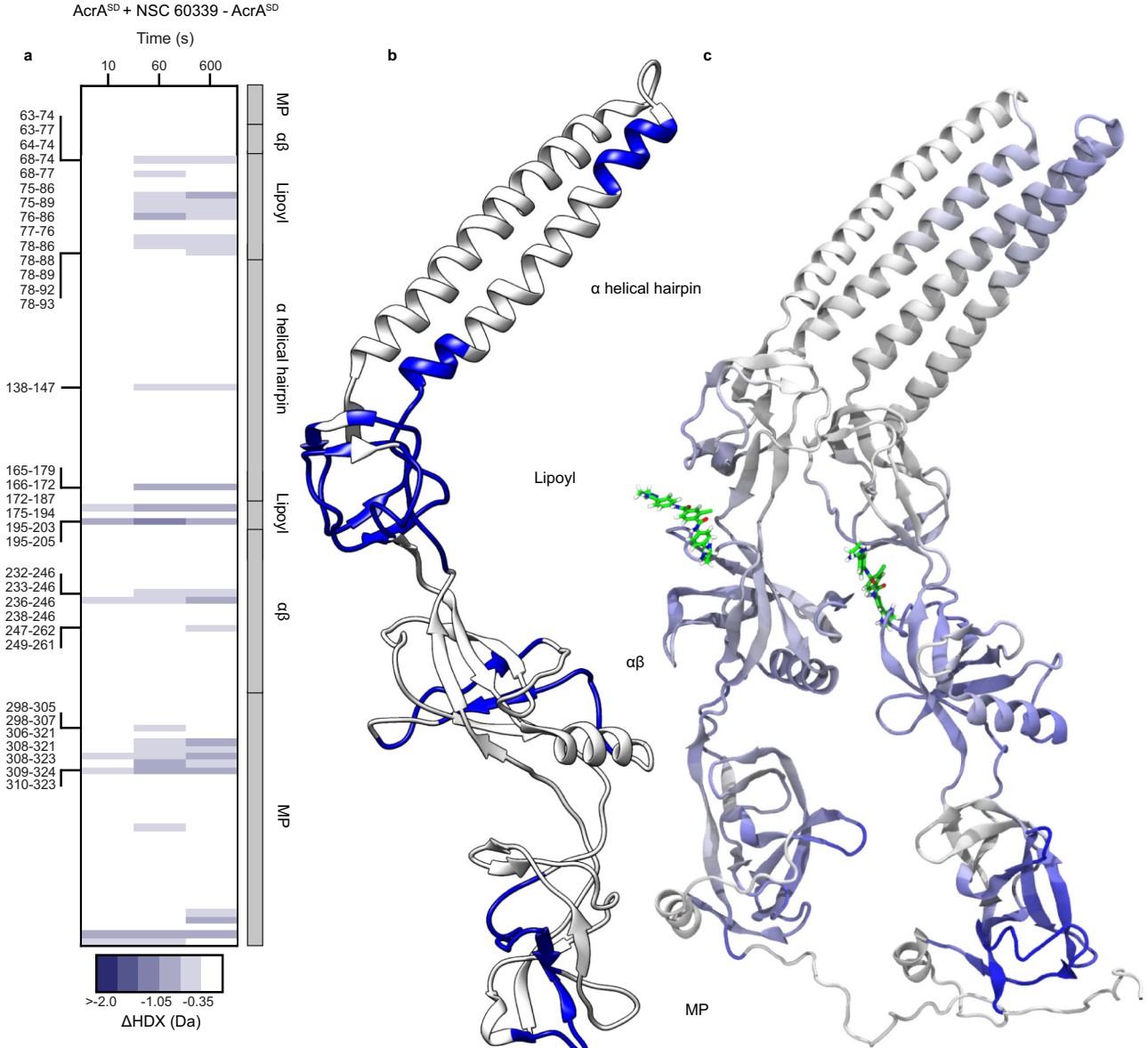

**Fig. 4 | The effect of NSC 60339 on AcrA$^{SD}$ structural dynamics. a** Chiclet plot displaying the differential HDX (ΔHDX) plots for AcrA$^{SD}$ +/− NSC 60339 for all time points collected. Blue signifies areas with decreased HDX between states. We defined significance as a ± ≥ 0.35 Da change (see 'Methods'), with a *P* value ≤ 0.01 in a two-sided Welch's *t* test (7 independent measurements: $n_{biological}$ = 2 and $n_{technical}$ = 3-4). White areas represent regions with insignificant ΔHDX. Source data are provided as a Source data file. **b** ΔHDX for ((AcrA$^{SD}$ + NSC 60339) − AcrA$^{SD}$) for the 10 min time point is painted onto the AcrA structure (PDB:5O66) using HDeXplosion and Chimera[23,55,56]. **c** Simulated AcrA$^{SD}$ coloured according to the difference in RMSF over the last 70 ns of 100-ns simulations of the bound and apo states, averaged over four replicas for each. Blue indicates that the RMSF is greater in the apo state than the bound state while white indicates they are similar in both states (the colour range is from −2 Å, blue, to 0 Å, white).

(Fig. 5b and Supplementary Fig. 15). Hence, only these Cys residues of AcrA are vulnerable to MTS-R6G binding and inhibition, whereas other Cys residues are either inaccessible in the AcrAB-TolC complex or binding of MTS in these sites does not affect the function of the complex. None of these residues are implicated in the characterised AcrA:AcrB binding sites, so the inhibitory effect on efflux is not due to perturbed interactions in the assembly of AcrAB-TolC[44]. One possibility for this inhibitory effect is that MTS binding causes the reduction in AcrA conformational flexibility, but definite conclusions on restricted dynamics cannot be drawn here. However, the results show that these three sites within the flexible hinges between AcrA domains are critical for efflux of the AcrAB-TolC pump.

T205 is in site IV, the proposed binding site of NSC 60339, and further evidences this area is a druggable site for AcrA inhibition.

Interestingly, two other residues were mutated in site IV, yet had no effect on efflux (R183 and D284). These residues reside on the periphery of site IV (Fig. 5b) and are likely less critical for AcrA function than T205, which in the core of the cleft. Furthermore, T205 showed significant protection in the presence of NSC 60339 in both AcrA constructs, highlighting its importance for AcrA flexibility. The two other residues that had an inhibitory effect on efflux are located away from site IV, at the flexible linker between the αβ barrel-MP domains. N232C and L50C are both located at the bottom of this region and orientated in the same direction (Fig. 5c). The other residues that had no effect on efflux were positioned above N232 and L50 and orientated in the opposite direction. Interestingly, both positions were also found in significantly protected peptides in the presence of NSC 60339 across our HDX-MS experiments. This finding suggests NSC 60339 is

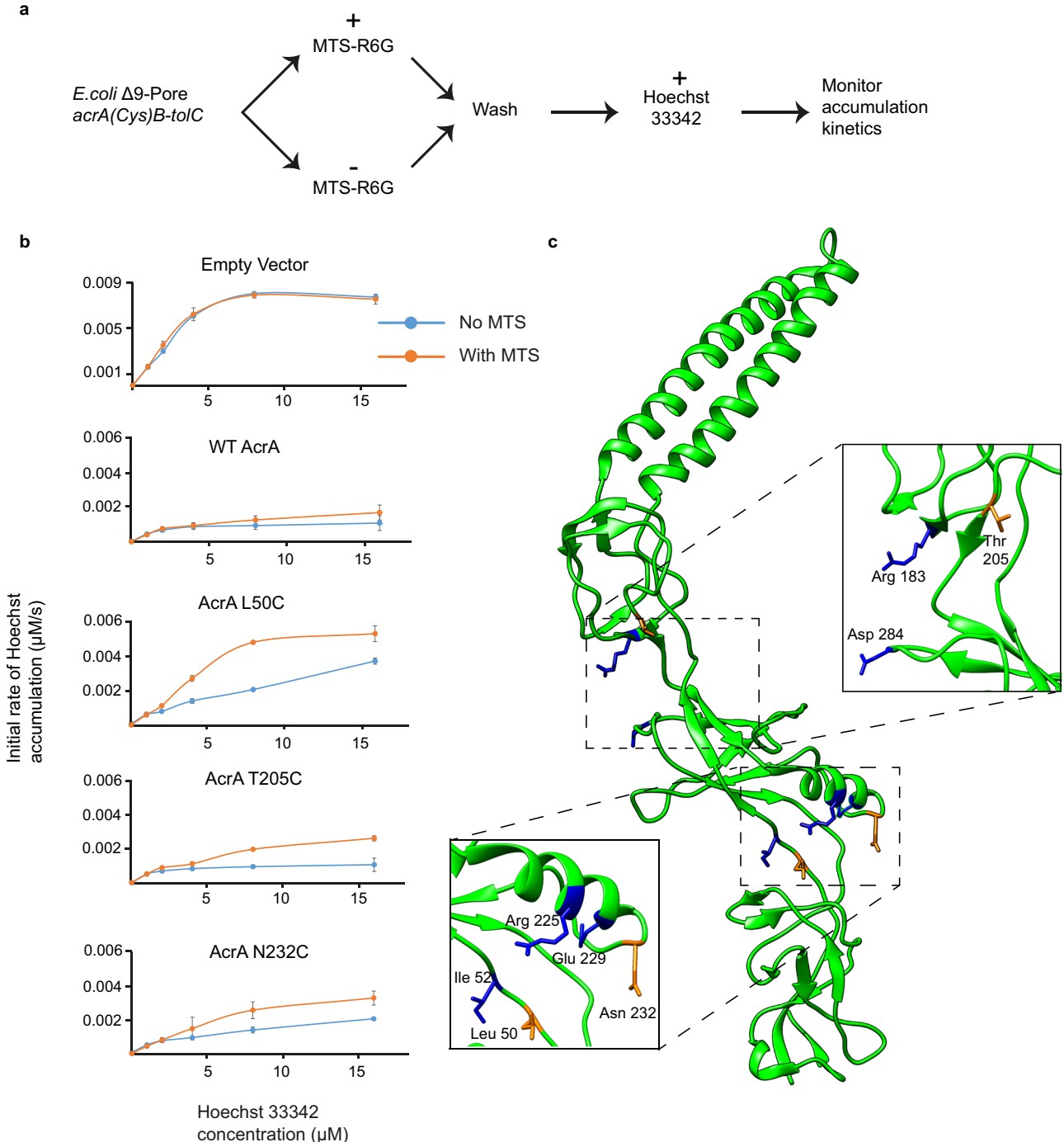

**Fig. 5 | The effect of Cys-reactive MTS probe on the efficiency of AcrAB-TolC.**
**a** Covalent efflux inhibition assay experimental process. MTS-R6G is the Cys reactive probe and Hoechst 33342 is the fluorescent efflux substrate. **b** *E. coli* Δ9-Pore cells producing AcrAB-TolC complex carrying the indicated AcrA variants were split into two aliquots and one of the aliquots was treated with a Cys-reactive probe MTS. After incubation for 15 min at 37 °C, cells were washed and the intracellular accumulation of Hoechst was analysed as described previously[61]. Kinetic data were fitted into a burst-single exponential decay function and the calculated initial rates of Hoechst accumulation (µM/s) are plotted as a function of the externally added concentration of Hoechst[61]. Plots are the mean and error bars indicate standard deviation (*n* = 3) from independent measurements. Source data provided as Source data file. **c** AcrA structure (PDB 5O66) with mutated residues highlighted[23]. Blue residues have no effect on efflux and orange residues had an effect.

stabilising this area downstream from its binding site. Therefore, this provides another druggable site on AcrA for the future design of EPIs.

## Discussion

In summary, our findings support that NSC 60339 positions itself in the cleft between the lipoyl and αβ barrel domains (site IV) and causes long-range restriction in backbone dynamics of AcrA across all four of its domains. This behaviour was observed in both monomer and pseudo-dimer constructs. We propose NSC 60339 acts as a molecular wedge within this cleft, significantly restricting the structural dynamics of AcrA, which could have implications for the conformational transitions required during the functional rotation of the AcrAB-TolC pump (Fig. 6). A recent study of the assembled AcrAB-TolC efflux pump in situ confirms that one of the AcrA protomers in the AcrA

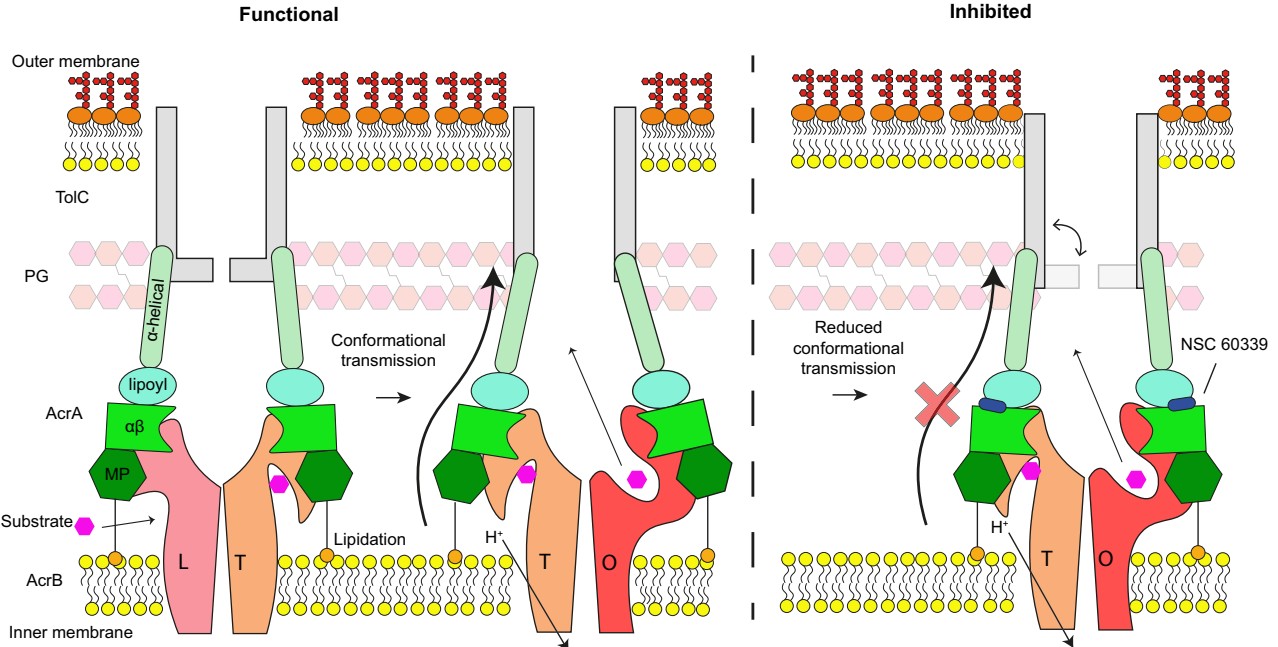

**Fig. 6 | Schematic of AcrAB-TolC inhibition by NSC 60339.** Under normal conditions, as AcrB cycles through its three states of its rotational mechanism (L Loose, T Tight, O Open) the conformational transition information is transmitted through AcrA to TolC, ensuring it is in the 'open' state for efflux. AcrA becomes conformationally restricted once NSC 60339 'wedges' between its lipoyl and αβ barrel domains, reducing its ability to transmit the conformational movements of AcrB and, subsequently, inhibiting functional rotation and efflux. Schematic based on in situ structure reported in Chen et al.[45].

dimer is anchored to the inner membrane and interacts with AcrB, while also stretching to interact with the peptidoglycan layer and TolC[45]. The N-terminal region of the second protomer is suggested to interact with the AcrB PC2 subdomain, which undergoes extensive conformational changes throughout the rotational mechanism[23]. This architecture could allow AcrA to communicate conformational changes in AcrB, to TolC, as it cycles through the rotational mechanism, resulting in TolC transitioning to the 'open' state. Thus, NSC 60339 inhibition may result in AcrA losing its ability to accommodate changes across the periplasm and communicate the conformational changes of AcrB to TolC, required for efflux. This could affect the pump in several ways; (1) The interaction of AcrA with AcrB and TolC may become disrupted as AcrA becomes more rigid, (2) AcrA may not be able to maintain a sealed channel during functional rotation, and 3) TolC may not be 'opened' as efficiently.

Recent work has identified AcrA as a necrosignal[15,16]. Dead cells release AcrA, which binds to TolC in the outer membrane of live cells to send a signal, increasing drug efflux and the expression of other pumps. It has been shown that NSC 60339 can inhibit both efflux and necrosignalling in AcrA[16]. Furthermore, while novobiocin does not inhibit efflux, as its primary target in bacteria is DNA Gyrase, it can also inhibit necrosignalling, albeit not to the same degree as NSC 60339[46]. It is proposed that AcrA interacts with TolC via its α-helical hairpin contacting the TolC pore. We show that a global stabilisation of AcrA by NSC 60339 may explain its necrosignalling inhibition ability, possibly by reducing its ability to bind to TolC on the outside of cells. However, novobiocin has also been shown to inhibit necrosignalling in AcrA, and we did not observe a significant effect of its binding on AcrA structural dynamics. Therefore, novobiocin may inhibit necrosignalling in a different fashion, or the structural dynamics may only form part of AcrA's necrosignalling ability. The necrosignal function of AcrA has only recently been discovered and, therefore, further investigations into the mechanisms of necrosignalling are required to understand this ability, and how best to inhibit it to combat bacterial multidrug resistance generation via this process.

Overall, in this work we propose a mechanism of action for the AcrA inhibitor NSC 60339. This provides a model for targeting the AcrAB-TolC multidrug efflux pump, providing a platform to understand and develop the next generation of EPIs. AcrA has been seen to exhibit interchangeability with other PAPs in non-cognate systems, highlighting its flexibility[47]. Therefore, uncovering the structural dynamics of AcrA is essential to understanding its role as a flexible PAP within an array of assembled multidrug efflux complexes. More generally, we show how HDX-MS can be used in combination with MD simulations, high-resolution structural information and microbiology assays to understand these dynamics and to determine the molecular mechanisms of inhibition, as previously seen with AcrB[48].

## Methods

### Cloning, expression, and purification

**AcrA[S].** AcrA[S] lacking the signal peptide residues 1–24, and the site of lipidation Cys25, was cloned in a pET28a plasmid with a 6xHis and LE linker (all primers used for cloning can be found in Supplementary Table 4)[17,27]. AcrA[S] is purified from the cytoplasmic fraction as previously reported[17]. Briefly, pET28a containing AcrA[S] was transformed into *E. coli* C43(DE3) *ΔacrAB* cells. 7 ml of an overnight Luria-Bertani (LB) culture was added to 1 L of pre-warmed LB culture (x6) containing 30 μg/ml kanamycin and grown at 37 °C until an $OD_{600}$ of 0.6 was reached. The culture was induced with 1 mM IPTG and grown for 3 h at 37 °C. The cells were harvested by centrifugation at $4200 \times g$ for 30 min and washed with ice cold phosphate buffer saline (PBS).

Cell pellets were frozen at −20 °C, and then thawed and resuspended in Buffer A (50 mM sodium phosphate, 150 mM NaCl, 5 mM imidazole, pH 7.4) and supplemented with a protease inhibitor (Roche), 100 mM PMSF, 2 μl Benzonase, 5 mM β-ME and lysozyme. The cell suspension was lysed by sonication at an amplitude of 40, $6 \times 15$ s with 90 s breaks. Insoluble material was removed by centrifugation at $20,000 \times g$ for 30 min at 4 °C. Membranes were pelleted at $200,000 \times g$ for 1 h at 4 °C.

The supernatant containing the cytosolic fraction was collected and filtered through a 0.22 μm filter (Thermo Fisher Scientific). The

sample was loaded onto a 1 ml HiTrap column (GE Healthcare) equilibrated in Buffer B (50 mM sodium phosphate, 300 mM NaCl, 20 mM imidazole, 10% Glycerol, pH 7.4). The column was washed with 20 column volumes (CV) of Buffer B, 10 CV of Buffer B with 50 mM imidazole, and eluted with Buffer B containing 500 mM imidazole. AcrA$^S$ was directly loaded onto a HiLoad 16/600 Superdex 200 pg size exclusion chromatography (SEC) column (GE Healthcare) equilibrated in Buffer C (50 mM sodium phosphate, 150 mM NaCl, 10% Glycerol, pH 7.4). A 1 ml min$^{-1}$ flow rate was used for the HiTrap and SEC column. Peak fractions of AcrA$^S$ were pooled and flash frozen at −80 °C. SDS-PAGE and nMS was used to assess purity and concentration was determined via Nanodrop.

**AcrA$^L$.** AcrA$^L$ containing the full length AcrA was amplified from *E. coli* genomic DNA and cloned into a pET28a plasmid using *NcoI* and *XhoI* restriction enzymes, with an LE linker and 6xHis. Cell expression, growth, and lysis was the same as AcrA$^S$.

Membranes were pelleted at 200,000 × *g* for 1 h at 4 °C. Membrane pellets were resuspended to 40 mg/ml in ice cold Buffer A supplemented with 100 mM PMSF and a protease inhibitor tablet and homogenised with a Potter-Elvehjem Teflon pestle and glass tube. AcrA$^L$ was solubilised from homogenised membranes with 1% (w/v) n-dodecyl-β-D-maltoside (DDM) detergent (Anatrace) at 4 °C. Insoluble material was removed by centrifugation at 100,000 × *g* for 30 min at 4 °C. The HiTrap and SEC stages are the same as above, except the buffers all contained 0.03% (w/v) DDM.

**AcrA$^{SD}$.** AcrA$^{SD}$ containing one AcrA$^S$ molecule with a Cys25Ala mutation, a TRRIT linker, and a second AcrA$^S$ molecule lacking residues 1–25, and containing an LE linker and 6xHis was cloned into a pET21d+ plasmid using *NcoI* and *XhoI* restriction enzymes. For ease of screening potential positives, a *BamHI* site was engineered into the linker region. Cell expression, growth, and purification was identical to AcrA$^S$ except 100 µg/ml ampicillin was used in the cell growth.

**AcrB.** AcrB was expressed and purified as previously reported[48]. Briefly, pET15b plasmid containing AcrB was transformed into *E.coli* C43(DE3) *ΔacrAB* cells. 7 ml of an overnight LB culture was added to 1 L of pre-warmed LB culture (×6) containing 100 µg/ml ampicillin and grown at 37 °C until an OD$_{600}$ of 0.6-0.8 was reached. The culture was induced with 1 mM IPTG and grown for 16–18 h at 18 °C. The cells were harvested by centrifugation at 4200 × *g* for 30 min and washed with ice cold PBS.

The cell pellet was resuspended in Buffer A (50 mM sodium phosphate, 150 mM NaCl, pH 7.4) and supplemented with a protease inhibitor (Roche), 100 µM PMSF, 2 µl Benzonase and 5 mM β-ME. The cell suspension was passed twice through a microfluidizer processor (Microfluidics) at 25,000 psi and 4 °C. Insoluble material was removed by centrifugation at 20,000 × *g* for 30 min at 4 °C. Membranes were pelleted at 200,000 × *g* for 1 h at 4 °C (Thermo Fisher Scientific). Membrane pellets were resuspended to 40 mg/ml in ice cold Buffer A supplemented with 100 µM PMSF and a protease inhibitor tablet and homogenised with a Potter-Elvehjem Teflon pestle and glass tube.

AcrB was extracted from the homogenised membranes with 2.5% (w/v) SMA 2000 copolymer for 2 h at room temperature to form native nanodiscs. Insoluble material was removed by centrifugation at 100,000 × *g* for 30 min at 4 °C. The sample was incubated with 1 ml super nickel NTA agarose affinity resin (Generon) equilibrated in Buffer B (50 mM sodium phosphate, 300 mM NaCl, 20 mM imidazole, 10% Glycerol, pH 7.4) overnight at 4 °C with gentle agitation. The sample was transferred to a gravity flow column and washed with 20 CV of Buffer B, 10 CV of Buffer B with 50 mM imidazole, and eluted with Buffer B containing 500 mM imidazole. AcrB was then buffer exchanged to Buffer C (50 mM sodium phosphate, 150 mM NaCl, 10% Glycerol, pH 7.4) using a PD-10 desalting column (GE Healthcare) and flash

frozen at −80 °C. SDS-PAGE was used to assess purify and concentration was determined via Nanodrop.

## Native mass spectrometry
Purified AcrA$^S$, AcrA$^{SD}$, AcrA$^L$ were exchanged into a volatile solution (100 mM ammonium acetate, pH 6.0/7.4) +/−0.03% DDM using a centrifugal exchange device (Micro Bio-Spin 6, Bio-Rad) according to the manufacturer's instructions, or via a SEC Superdex 10/200 Increase column at a 0.4 ml/min flow rate. Native mass spectrometry experiments were performed on a Synapt G2-Si mass spectrometer (Waters), or The Q-Exactive Plus UHMR (Thermo Fisher Scientific).

Each sample was loaded into homemade gold-coated borosilicate glass capillaries and mounted onto a Synapt G2-Si mass spectrometer (Waters), where native mass spectrometry experiments were performed. Nano-electrospray ionisation (nESI) was performed, and generated protein ions were drawn into the vacuum of the mass spectrometer. The following instrument parameters were carefully optimised to avoid ion activation and protein unfolding: capillary voltage: 1.4 kV, sampling cone: 30 V, trap DC bias: 15 V, trap collision energy: 2 V, and transfer collision energy: 2 V. Pressures were set to 5.91 × 10$^{-2}$ mbar in the source region (backing) and to 1.58 × 10$^{-2}$ mbar in both trap and transfer collision cells (collision gas: He).

For AcrA$^L$, a higher energy was required to release the protein from its hydrophobic environment. The sampling cone therefore was set to 120 V and the trap collision energy 50-200 V. The other instrument parameters were kept the same.

Only AcrA$^{S/SD}$ were measured using The Q-Exactive Plus UHMR (Thermo Fisher Scientific). With samples containing novobiocin, the drug was added to the protein and left to incubate for 30 min prior to MS analysis. The UHMR settings used were: 1.5 kV spray voltage, capillary temperature 60 °C, IST off, Extended trapping 60.

Data were processed and analysed using MassLynx v.4.1 (Waters) and UniDec (v.4.2.2)[49].

## Preparation of ligands for hydrogen deuterium exchange mass spectrometry
NSC 60339 and novobiocin was purchased from MedChemExpress and Caymen Chemical, respectively. Stock solutions of NSC 60339 (10 mM) and novobiocin (600 µM) were made in 100% DMSO and sonicated for 2–3 h to ensure solubility. Obtaining a maximal percentage of protein:drug complex during deuterium labelling is an important consideration for HDX-MS experiments. NSC 60339 and novobiocin both bind to AcrA$^S$ with µM affinity (NSC 60339 with a $K_D$ of 78 µM and novobiocin with a $K_D$ of 4.3 µM, as measured by previous SPR measurements at pH 6.0)[10]. To ensure we had maximum protein:drug complex present in our labelling conditions, we used the equation below[50]:

$$\text{Fraction of bound protein} = \frac{(L_T + P_T + K_D) - \sqrt{(L_T + P_T + K_D)^2 - 4L_T P_T}}{2P_T} \quad (1)$$

Protein was incubated with 500 µM NSC 60339 or 30 µM of novobiocin before being diluted into labelling buffer containing the same concentration of drug. 5% DMSO was kept consistent throughout experimentation to ensure drug solubility.

## Hydrogen deuterium exchange mass spectrometry
HDX-MS experiments were performed on a nanoAcquity ultra-performance liquid chromatography (UPLC) Xevo G2-XS QTof mass spectrometer system (Waters). Optimised peptide identification and peptide coverage for AcrA$^{S/SD}$ was performed from undeuterated controls. The optimal sample workflow for HDX-MS of AcrA$^{S/SD}$ was as follows: 5 µl of AcrA$^{S/SD}$ (20 µM) was diluted into 95 µl of either equilibration buffer (50 mM sodium phosphate, 150 mM NaCl, 5% DMSO, +/−NSC 60339/Novobiocin, pH 6.0) or labelling buffer (deuterated

equilibration buffer) at 20 °C. After fixed times of deuterium labelling, the samples were mixed with 100 μl of quench buffer (formic acid, 1.6 M GuHCl, 0.1% Fos-choline, pH 1.9) to provide a quenched sample at pH 2.4. 70 μl of quenched sample was then loaded onto a 50 μl sample loop before being injected onto an online Enzymate™ pepsin digestion column (Waters) in 0.1% formic acid in water (200 μl/min flow rate) at 20 °C. The peptic fragments were trapped onto an Acquity BEH c18 1.7 μM VANGUARD pre-column (Waters) for 3 min. The peptic fragments were then eluted using an 8–35% gradient of 0.1% formic acid in acetonitrile (40 μl/min flow rate) into a chilled Acquity UPLC BEH C18 1.7 μM 1.0 × 100 mm column (Waters). The trap and UPLC were both maintained at 0 °C. The eluted peptides were ionised by electrospray into the Xevo G2-XS QTof mass spectrometer. $MS^E$ data were acquired with a 20–30 V trap collision energy ramp for high-energy acquisition of product ions. Argon was used as the trap collision gas at a flow rate of 2 ml/min. Leucine enkephalin was used for lock mass accuracy correction and the mass spectrometer was calibrated with sodium iodide. The online Enzymate™ pepsin digestion column (Waters) was washed three times with pepsin wash (1.5 Gu-HCl, 4% MeOH, 0.8% formic acid, 0.1% Fos-choline), and a blank run was performed between each sample with pepsin wash without Fos-choline, to prevent significant peptide carryover between runs.

All deuterium time points and controls were performed in triplicate/quadruplicate. Sequence identification was performed from $MS^E$ data of digested undeuterated samples of $AcrA^{S/SD}$ using ProteinLynx Global Server 3.0.2 software (Waters). The output peptides were then filtered using DynamX (v. 3.0) using these parameters: minimum intensity of 1481, minimum and maximum peptide sequence length of 5 and 20 respectively, minimum MS/MS products of 1, minimum products per amino acid of 0.11, and a maximum MH+ error threshold of 5 ppm[51]. All the spectra were visually examined and only those with a suitable signal to noise ratio were used for analysis. The amount of relative deuterium uptake for each peptide was determined using DynamX (v. 3.0) and are not corrected for back exchange since only relative differences were used for analysis and interpretation and there was no benefit from normalising the data[52]. The only exception is for the HDX heatmap of $AcrA^S$, to which back exchange correction was applied (Supplementary Fig. 3). The relative fractional uptake (RFU) was calculated from the following equation, where $Y$ is the deuterium uptake for peptide $a$ at incubation time $t$, and $D$ is the percentage of deuterium in the final labelling solution:

$$RFU_a = \frac{Y_{a,t}}{MaxUptake_a \times D} \qquad (2)$$

A significance level cut off was decided for each dataset based on the quality of the data as previously defined[53,54]. Confidence intervals for differential HDX measurements of any individual time point were determined according to a two-sided Welch's $T$ test using HDeXplosion software[55]. Only peptides which satisfied the corresponding ΔHDX cutoff and a confidence interval of 99% were considered significant. All ΔHDX structure figures were generated from the data using HDeXplosion and Chimera[55,56]. All supporting data and meta-data are reported in the Source data file (Supplementary Figs. 3, 7, 9, 10, and 14).

### Maximally labelled control (MaxD)
5 μl of protein (+/−5% DMSO) was diluted in 95 μl labelling buffer (+/−5% DMSO) at labelled at pH 6.0 for 251 min at 50 °C. This is following the previously determined protocol of labelling for 10 min at pH 7.4 at -5 °C below the Tm of the protein[57]. This is to ensure maximum deuterium incorporation of unfolded protein. The deuterium content in the reaction mixture is identical to the corresponding HDX experiment. After 251 min, the proteins were left at room temperature

for 2 min, then on ice for 2 min before being flash frozen and stored at −80 °C until LC-MS analysis.

### SMA-PAGE native gels
Novex Tris-Glycine Gels (4-20%) were used for the native SMA-PAGE gels (Thermo Fisher Scientific). All native gels were run using Tris-Glycine running buffer (25 mM Tris pH 8.8, 192 mM Glycine). The native loading dye was Novex Tris-Glycine Native Sample Buffer (Thermo Fisher Scientific). SMA-PAGE gels were run at 150 V for 90–120 min at 4 °C. To detect protein, gels were stained with Quick Coomassie Stain (NeoBiotech) according to the manufacturers instructions.

### Surface plasmon resonance
$AcrA^S$ and $AcrA^{SD}$ protein were immobilised using the amine coupling method. For this purpose, CM5 chip (BiaCore) surfaces were activated with 0.05 M N-hydroxysuccinimide and 0.2 M N-ethyl-N-(3 -diethyla-minopropyl) carbodiimide. $AcrA^S$ and $AcrA^{SD}$ were injected over surfaces immediately after activation. After immobilisation, the excess of reactive groups was blocked by injecting 0.5 M ethanolamine HCl (pH 8.0). The immobilisation and subsequent binding experiments were conducted in running buffer containing 20 mM HEPES-KOH (pH 7.0), 150 mM NaCl, and 0.03% DDM supplemented with 5% DMSO. The CM5 chip contains four chambers, whereas the first (control surface) was activated and processed in the same way but the protein was omitted during the immobilisation step. The second and third chambers contained the immobilised $AcrA^S$ and $AcrA^{SD}$ (ligand). The immobilised densities of both proteins (ligand) were 4743 and 4054 response units (RU), respectively. The sensorgrams were collected and analysed as described before[58,59].

### Site-directed mutagenesis
All amino acid substitutions in *acrA* were introduced by QuickChange II XL Site-Directed Mutagenesis Kit using p151*acrAB*His as the template[60]. Primer design and Polymerase chain reaction (PCR) reaction for each substitution were performed by following manufacturer's protocol.

*E. coli* Δ9-Pore strain (*acrB ΔacrD ΔacrEF*::spc *ΔemrB ΔemrY ΔentS*::cam *ΔmacB ΔmdtC ΔmdtF att*Tn7::mini-Tn7T Tp[r] *araC* P_{araBAD} *fhuAΔC/Δ4L*) was constructed previously[43]. This strain was used in antibiotic susceptibility and covalent inhibition studies.

### Minimal inhibitory concentrations
Susceptibilities of the *E. coli* Δ9-Pore cells containing plasmid-borne AcrA(Cys)AcrB variants against SDS, Novobiocin, Erythromycin, and Vancomycin, were determined by two-fold broth microdilution. Briefly, overnight cultures were sub-cultured in LB broth (tryptone, 10 g/l; yeast extract, 5 g/l; NaCl, 5 g/l), and cells were grown at 37 °C in a shaker at 225 rpm until $OD_{600}$ reach to 0.2–0.3. For the proper expression of the pore, L-arabinose (final concentration of 0.1%) was added to each culture, and cells were further grown until $OD_{600}$ reached 1.0. The minimum inhibitory concentration of each bacterial strain against different antibiotics was measured in 96-well plates. Exponentially growing cells were added to each well and incubated for 18 h. Plates were scanned to determine the final $OD_{600}$ by Spark 10M microplate reader (TECAN).

### Covalent inhibition experiments
Overnight culture of *E. coli* Δ9-Pore cells carrying p151-AcrABHis plasmid with the indicated substitutions in AcrA were sub-cultured with 0.1% arabinose for 6 h to achieve an $OD_{600}$ of 1.0. For covalent inhibition of AcrA, cells were split into two aliquots and one aliquot was incubated with 20 μM MTS for 15 min at 37 °C. The comparator cell aliquot was incubated with a blank solvent (dimethyl sulfoxide) under the same conditions. After incubation, cells were washed twice with HMG buffer (50 mM HEPES-KOH buffer pH 7.0, 1 mM magnesium

sulfate and 0.4 mM glucose) and resuspended in HMG buffer to an $OD_{600}$ of ~1.0, at room temperature. Different concentrations of Hoechst were tested to measure the substrate efflux efficiency of each mutant cells by measuring kinetics of Hoechst accumulation at $\lambda ex = 350$ nm and $\lambda em = 450$ nm. Data were normalised and kinetic parameters were calculated as described previously using a MATLAB program[61]. Briefly, the time courses of Hoechst uptake (Supplementary Fig. 13) were fit to the burst-single exponential decay function $F = A_1 + A_2 \cdot (1-\exp(-kt))$, where $A_1$ and $A_2$ describe the magnitude of the fast and slow steps, respectively, and $k$ is the rate of the slow step. The fast and slow steps were attributed to Hoechst binding to the lipids (i.e. in the periplasm) and chromosomal DNA (i.e. in the cytoplasm), respectively. The initial rates for Hoechst accumulation in the cytoplasm were calculated as $V_1 = A_2 \cdot k$ and plotted in Fig. 5b and Supplementary Fig. 15.

## Circular dichroism (CD) spectroscopy
All CD spectra were measured in an Aviv Circular Dichroism Spectrophotometer, Model 410. A final protein concentration of 0.4/0.26 mg/ml was used in a quartz rectangular Suprasil demountable cell of 1.0 mm pathlength. Each sample was scanned twice from 260 to 190 nm at 1 nm intervals, with an averaging time of 0.5 s, at temperatures ranging from 25 to 90 °C, with 5 °C increments. DMSO was kept at 0.5%. Spectra analysed on SigmaPlot.

## Molecular dynamics simulations
MD simulations of the NSC 60339-bound monomer were initialised using the docked structure from Darzynkiewicz et al.[14], while those of the apo used the same structure with the compound removed. Each of the two systems was solvated in an ~(170-Å)³ cubic water box in order to allow for tumbling of the protein without using orientational restraints and ionised with 150 mM NaCl, resulting in a system size of ~476,000 atoms (Supplementary Fig. 16). The dimer system was constructed starting from the AcrA dimer structure in PDB 2F1M[18]. The MP domain was added based on the monomer structure, and residues in the lipoyl and αβ barrel domains were adjusted to match their positions in the NSC 60339-bound structure, after which the compound was modelled in to both copies of AcrA. Apo and bound systems were solvated in an ~(210-Å)³ cubic water box and ionised with 150 mM NaCl, resulting in a system size of ~906,000 atoms.

Each system was equilibrated for 0.5 ns with all protein and ligand atoms restrained, followed by 4.5 ns with only the protein backbone and ligand atoms restrained. Then, each system was equilibrated for 100 ns in triplicate. Simulations were run using either NAMD2.14[62] or NAMD3[63] depending on the computational resource used, and the CHARMM36m force field[64]. Force-field parameters for NSC-60339 were generated using the CGenFF webserver[65] and are provided as Supplementary Data 2. A time step of 2 fs was used, with short-range non-bonded interactions (cut off of 12 Å with a switching function starting at 10 Å) updated every time step and long-range electrostatics updated every other time step using the particle mesh Ewald method[66]. A constant temperature of 310 K was maintained using Langevin dynamics, while a constant pressure of 1 atm was maintained using a Langevin piston. All results presented were averaged over three replicas.

All system preparation and analysis were carried out using VMD. RMSF and SASA were measured using VMD's 'measure rmsf' and 'measure sasa' functions. In the case of RMSF, the fluctuations are measured using the average position over the sampled frames as a reference. The root-mean-square deviation (RMSD) for individual domains were tracked and are plotted in Supplementary Fig. 17 for the AcrA$^S$ and Supplementary Fig. 18 for AcrA$^{SD}$.

## Reporting summary
Further information on research design is available in the Nature Portfolio Reporting Summary linked to this article.

## Data availability
The HDX-MS data generated in this study have been provided in the Source data file, and the HDX-MS summary tables and uptake plots have been provided as Supplementary Data 1 and 3. Furthermore, the mass spectrometry proteomics data have been deposited to the ProteomeXchange Consortium via the PRIDE partner repository with the dataset identifier PXD041359. The MD simulations generated in this study have been provided as Supplementary Data 2. The protein structures from other publications referenced in this paper are accessible under the PDB accession codes 5O66 and 2F1M. The AlphaFold2 structure was generated from the UniProt entry P0AE06. Source data are provided with this paper.

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

## Acknowledgements

Work at King's College London was supported by a UKRI Future Leaders Fellowship (MR/S015426/1) to E.R. and a King's College London PhD studentship to B.R.L. This work was also supported by US National Institutes of Health grant R01-AI052293 to H.I.Z., J.M.P., and J.C.G. Computational resources were provided through XSEDE (TG-MCB130173), which is supported by the US National Science Foundation (NSF; ACI-1548562). This research used resources at the Compute and Data Environment for Science (CADES) at ORNL, which is managed by UT Battelle, LLC, for DOE under contract DE-AC05–00OR22725. This work also used the Hive cluster, which is supported by the NSF (1828187) and is managed by the Partnership for an Advanced Computing Environment (PACE) at GT. The Q-Exactive Plus UHMR at the University of Leeds used for native MS was funded by The Wellcome Trust (208385/Z/17/Z). A.J.H. was funded by a BBSRC IPA grant (BB/R018561/1/) in collaboration with GSK and UCB Pharma. The authors thank Valeria Calvaresi for advice with HDX analysis and statistics.

## Author contributions

B.R.L., J.C.G., H.I.Z., and E.R. designed the project; B.R.L., L.M.N.S., D.H., and E.R. performed all mass spectrometry experiments and analysis, except for UHMR measurements that were conducted by A.H. and F.S.; B.R.L., M.M., E.R., and H.I.Z. cloned, purified, and characterised all protein constructs. M.R.U. and H.I.Z. performed bacterial efflux and susceptibility assays and analysis; K.M.K., J.M.P., and J.C.G. carried out molecular docking and dynamics experiments and post-molecular dynamics analyses; F.S., J.M.P., D.H., J.C.G., H.I.Z., and E.R. supervised or financially supported the project; B.R.L., J.C.G., H.I.Z., and E.R. wrote the manuscript with input from the other authors.

## Competing interests

The authors declare no competing interests.
