## [Peer Review File · Nature Communications]

REVIEWER COMMENTS

Reviewer #1 (Remarks to the Author):

In this manuscript, the authors demonstrated a mechanism of inhibition against the periplasmic adaptor protein, AcrA of the AcrAB-TolC tripartite multidrug efflux system. Given the widespread nature of this efflux system in ESKAPE pathogens, understanding the drug efflux process and inhibition is of paramount for developing next-generation antimicrobial therapeutics. Here, the authors used a combination of hydrogen/deuterium exchange mass spectrometry, cellular efflux assays, and molecular dynamics simulations and defined the structural dynamics of AcrA in the presence of an inhibitor, NSC 60339 and a substrate, and proposed a model where NSC 60339 inhibits efflux by restricting AcrA structural dynamics, which could reduce the efficiency of AcrAB-TolC functional rotation. Overall, the results are of high quality and support the conclusions. The manuscript should be publishable in Nature Communications, but there are a few minor issues that should be addressed for the benefit of the reader, detailed below:

1. For the SD construct, the linking of one monomer to another is through TRVVT, but in Fig. 1C, it is confusing how it was shown T linking to another monomer and DDK left apart.
2. The observation that AcrA constructs exhibit multiple charge state distributions is very interesting. The authors attribute this to intrinsic disordered nature within the MP domain, which was previously unresolved in crystal structures before. Here, the authors could make use of Alphafold's structure and expand their discussion.
3. On page 6, in paragraph one, where the authors describe the AcrAS construct data, Fig. 1C was referred to instead of Fig. 1D.
4. Fig. S3 is called in the text before S2, I would suggest authors refer to the figures as the data appears in the text. Also, label X-axis in Fig S3.
5. Define the X-axis label HT in Fig. 5B.
6. To help the reader understand better, it is good to point out the exact zoom-in sections in Fig. 5C, just like how it is done in Fig 3.
7. On page 13, Fig. 1A is incorrectly cited, it should be Fig. 1C.
8. For consistency, please update the AcrA dimer in Fig. S11 to AcrASD.
9. Since the authors used native MS in this study, I wonder if they tried/managed to capture AcrB-AcrA interaction by native MS. This could help resolve the inconclusive interaction between AcrASD and AcrB that was shown in Fig. S11. I understand capturing membrane protein-soluble protein interaction by native MS is system dependent, but I am curious to know how this is in this case.

Reviewer #2 (Remarks to the Author):

This manuscript includes an ambitious number of technically challenging experiments to determine the mechanism of inhibition of the bacterial efflux pump Acr by a compound referred to as NSC60339. This efflux system plays a significant role in antibiotic resistance so new inhibitors could have a major impact on antibiotic-based therapy. The studies utilize a combination of H/DX MS, native MS, Molecular Dynamics simulations and functional export assays with several engineered variants of the wild type.

The H/DX results are 'clean' and straightforward in terms of interpretation. The NSC 60339 inhibitor affects specific regions throughout the length of the protein, whereas the more thoroughly studied novobiocin has no effect. Together the H/DX results demonstrate decreased H/DX which are consistent with global changes in hydrogen bonding spread throughout 'hotspots' in the ACRA. One technical issue that should be clarified is the fact that the exchange data are not corrected for back exchange. Why not? The system looks well behaved so the back exchanges should be accessible.

I am not an expert in MD simulations. There is an ongoing debate about whether MD and H/DX should correlate since the time scales are so dramatically different. When they don't correlate, this difference is invoked as an explanation but in this case they correlate remarkably well. So the local dynamics revealed by MD seem to report on the lower frequency dynamics that the H/DX reports on, in this case. Most important to the conclusions of the paper, the ligand-induced changes in dynamics are transmitted to sites remote from the binding site.

For the experiments described in Figure 4, with the engineered dimer AcrASD, the recovered values for K_D , k_{on} , and k_{off} should be reported. Was the K_D determined from the kinetic parameters or from the equilibrium RU plateaus shown in the Figure, or both? In fact the authors are using SPR in a 'nontraditional way to get stoichiometry information about binding which comes from the equilibrium RU values. What is the actual calculated stoichiometry based on this?

For the comparison of the H/DX to the MD simulations, the authors refer in the figure legend 4 to 'red' regions of decreased protection upon binding - however, there are no red regions, correct? Should the reference to red regions be left out? Also, the authors invoke SAS and show some red

regions of increase SAS, but this is not consistent with the H/DX in that region. Do the authors have an explanation?

For Figure 5, concerning the functional transport properties of the covalently adducted mutants, the axes labels are confusing. It looks like the reported parameter is rate of Hoescht accumulation rather than rate of efflux, so it's inversely related to AcrA function. That is OK, but it took a little effort to figure out what was plotted. So, please explicitly report the rates as micromolar accumulated/sec. Also, related to this, it isn't clear how the values on the y-axis are obtained. The authors refer to fitting the fluorescence or absorbance of Hoescht to a double exponential. Why do the kinetics behave as a double exponential, and which of the two rate constants is used in the plots in Figure 5, and why? How does the other rate constant behave? In short, it seems like a lot has been done to the raw data to get the plots and more explanation of this analysis is required.

In general, the results described with the cell-based efflux, or lack of accumulation, don't really report on any dynamic properties of AcrA. The authors are trying to make a correlation between drug-dependent loss of flexibility and loss of function but these assays don't probe dynamics. I believe that covalent adduction in the linker regions could prevent conformational changes required for function but the data don't reveal any change in dynamics, as the H/DX results do with the NSC60339. Interestingly the adducts must be fluorescent and fluorescence polarization might be exploited as a direct readout of dynamics, although I admit I'm not sure what the control sample would be. Alternatively, why not do H/DX on the 6RG-adducted proteins to see how the dynamics are affected. At the very least, the discussion should be rewritten to minimize the connection between the presumed effect of adduction on dynamics and function and just conclude that intact structure in the two linker regions is critical for function.

Minor points Throughout the manuscript, the word 'data' is used with singular grammar but 'data' is plural. Data 'are' analyzed and 'they' report on events. This should be corrected.

It would be useful to have the chemical structure of NSC60338 shown somewhere, in addition to the model of it in the docked complex with AcrA.

Reviewer #3 (Remarks to the Author):

In this study, the authors investigated the molecular mechanism of an inhibitor, NSC 60339, against AcrA using a combination of hydrogen/deuterium exchange mass spectrometry, cellular efflux assays, and molecular dynamics (MD) simulations. Their results demonstrate that the inhibitor could restrict the flexibility of AcrA, by forming a molecular wedge between the lipoyl and $\alpha\beta$ domains. I have some questions and suggestions, focusing on the MD simulations.

1. More details of the Method about molecular dynamics simulations should be provided.

The system size of ~476000/906000 atoms seems to be much larger than that in references 14 and 25. What are the compositions, and why is the large system size required? A detailed figure of the simulation systems should be given in the main text or in the supporting information.

How were the force field parameters of NSC 60339 generated?

Did the MD simulations performed at the same pH as the experiments? What are the protonation states of the titratable residues? The protonation states of titratable residues regulate the conformational changes of AcrA (ref 25).

How were the RMSFs calculated? What is the reference structure?

2. The convergence of the MD simulations should be provided.

The initial structures are from molecular docking or molecular modeling, and may need time to relax. What is more, the main results are the differences of RMSFs between the bound and apo states, the convergence of the MD simulations is essential for comparing different simulations. Therefore, RMSDs and RMSFs of each simulation should be shown in the supporting information at least. And it seems not a good idea to calculate the RMSFs based on the 0-100 ns trajectories.

3. Figure 3 is hard to read. The residues, F81 and F254, should be highlighted and labeled in panel B, as well as residues 308-324 in panel C. A color bar should be provided.

The difference is hard to tell from the colors, maybe the values should be given.

4. What about the comparison between the results of MD simulations and experiments? The effect from MS results (Figs. 2C and 4C) on AcrA structural dynamics looks different from that from MD simulations (Figs. 3C and 4D).

5. Page 8. "However, in MD simulations, both show reduced fluctuations (Fig. 3C) as well as reduced solvent-accessible surface area (SASA; Fig. 3D)," what is the "both" refer to?

6. Page 10. "It has also been shown that AcrA lacking the MP domain is still bound by NSC 60339, advocating that its stabilisation is allosteric to NSC 60339 binding to site IV." Is a reference missed in this sentence?

REVIEWER COMMENTS

Reviewer #1 (Remarks to the Author):

In this manuscript, the authors demonstrated a mechanism of inhibition against the periplasmic adaptor protein, AcrA of the AcrAB-TolC tripartite multidrug efflux system. Given the widespread nature of this efflux system in ESKAPE pathogens, understanding the drug efflux process and inhibition is of paramount for developing next-generation antimicrobial therapeutics. Here, the authors used a combination of hydrogen/deuterium exchange mass spectrometry, cellular efflux assays, and molecular dynamics simulations and defined the structural dynamics of AcrA in the presence of an inhibitor, NSC 60339 and a substrate, and proposed a model where NSC 60339 inhibits efflux by restricting AcrA structural dynamics, which could reduce the efficiency of AcrAB-TolC functional rotation. Overall, the results are of high quality and support the conclusions. The manuscript should be publishable in Nature Communications, but there are a few minor issues that should be addressed for the benefit of the reader, detailed below:

1. For the SD construct, the linking of one monomer to another is through TRVVT, but in Fig. 1C, it is confusing how it was shown T linking to another monomer and DDK left apart.

We thank the reviewer for pointing this out. This has now been amended in Figure 1C to show the correct linkage of Monomer 1-TRVVT-Monomer 2.

2. The observation that AcrA constructs exhibit multiple charge state distributions is very interesting. The authors attribute this to intrinsic disordered nature within the MP domain, which was previously unresolved in crystal structures before. Here, the authors could make use of AlphaFold's structure and expand their discussion.

The reviewer makes an excellent suggestion to use AlphaFold2 to expand our discussion. We have added the following text (page 7) and supplementary figure (Fig. S4) concerning our AlphaFold2 analysis and its insight into the intrinsic disorder within AcrA:

'Expanding on this notion further, we evaluated the structure of AcrA as predicted by AlphaFold2 (Fig. S4).^{37,38} AlphaFold2 provides a per-residue confidence score (pLDDT) between 0-100 for each residue; regions with a score of <50 may be unstructured. Regions 1-36 and 379-397 are both in the MP domain and contain many residues with a pLDDT score <50. This is in agreement with our mass spectrometry results that the MP domain contains unstructured regions'

Figure S4. AlphaFold2 prediction of AcrA structure. A. Predicted structure of AcrA. Colour coded based on the per-residue confidence score (pLDDT). A very low confidence score (<50) can indicate unstructured regions. B. Predicted aligned error plot. The colour position at (x, y) indicates AlphaFold2's expected position error at residue X, when the predicted and true structures are aligned on residue y. The colour bar represents how confident AlphaFold2's prediction is – dark green is high confidence, light green is low confidence.^{37,38}

3. On page 6, in paragraph one, where the authors describe the AcrAS construct data, Fig. 1C was referred to instead of Fig. 1D.

This has been corrected.

4. Fig. S3 is called in the text before S2, I would suggest authors refer to the figures as the data appears in the text. Also, label X-axis in Fig S3.

Order of the SI figures and x-axis label have now been corrected.

5. Define the X-axis label HT in Fig. 5B.

The X-axis of Figure 5B has been amended to define HT. The Y-axis labels have also been altered to "Initial rates of HT accumulation (mM/s)" to better define them.

6. To help the reader understand better, it is good to point out the exact zoom-in sections in Fig. 5C, just like how it is done in Fig 3.

This has been corrected.

7. On page 13, Fig. 1A is incorrectly cited, it should be Fig. 1C.

This has been corrected.

8. For consistency, please update the AcrA dimer in Fig. S11 to AcrASD.

This has been corrected.

9. Since the authors used native MS in this study, I wonder if they tried/managed to capture AcrB-AcrA interaction by native MS. This could help resolve the inconclusive interaction between AcrASD and AcrB that was shown in Fig. S11. I understand capturing membrane protein-soluble protein interaction by native MS is system dependent, but I am curious to know how this is in this case.

This is a very interesting experiment and something which we have attempted. Unfortunately, we have not been able to capture the correct bipartite or tripartite complex of AcrB₃-AcrA₆ or AcrB₃-AcrA₆-TolC₃ in the gas phase for Native MS analysis. This is likely due to technical limitations in getting such a large and dynamic membrane protein complex to ionize and remain intact in vacuum. Indeed, the only known success was achieved very recently by the group of Prof Carol Robinson who invented bespoke instrumentation and workflows to capture the complex from membrane extracts – indeed, even with these innovations the complex could still only be captured in an atypical AcrB₃-AcrA-TolC stoichiometry (doi: 10.1126/science.aau0976). We will continue to pursue these experimental avenues but at this time we are unable to explore these interactions in this detail using our current Native MS instrumentation.

Reviewer #2 (Remarks to the Author):

This manuscript includes an ambitious number of technically challenging experiments to determine the mechanism of inhibition of the bacterial efflux pump Acr by a compound referred to as NSC60339. This efflux system plays a significant role in antibiotic resistance so new inhibitors could have a major impact on antibiotic-based therapy. The studies utilize a combination of H/DX MS, native MS, Molecular Dynamics simulations and functional export assays with several engineered variants of the wild type.

The H/DX results are 'clean' and straightforward in terms of interpretation. The NSC 60339 inhibitor affects specific regions throughout the length of the protein, whereas the more thoroughly studied novobiocin has no effect. Together the H/DX results demonstrate decreased H/DX which are consistent with global changes in hydrogen bonding spread throughout 'hotspots' in the ACRA. One technical issue that should be clarified is the fact that the exchange data are not corrected for back exchange. Why not? The system looks well behaved so the back exchanges should be accessible.

We have performed a maximally deuterium labelled control (MaxD) and used it to calculate back exchange levels throughout our system which are reported in the HDX summary tables and MaxD protocol details in the revised Methods section.

The focus of our HDX-MS studies was to observe the effect of inhibitor binding on AcrA dynamics. To do this we performed differential HDX where the difference between two states (+/- drug) is assessed, because of this then we have not corrected for back exchange since only relative differences were used for analysis and interpretation and there was no benefit from normalizing the data. Nonetheless, we have included the MaxD data in the Source Data and in our PRIDE repository dataset for those who would like to correct this data for future modelling or data mining purposes.

A MaxD measure can be very useful for defining areas of intrinsic disorder in a protein, as areas which take up maximal deuteration at the earliest timepoints are indicative of a region which has no measurable secondary structure. We applied a MaxD normalization to our HDX heat map of AcrA^S in Fig. S3 and further validated the regions we identified as contributing to the disorder within AcrA. Note: that the previous heat map was constructed using DynamX (Waters) software, but this could not be used to perform a MaxD normalization, instead we used PyHDX software⁶⁹ which reports the same heat map but in a different format.

I am not an expert in MD simulations. There is an ongoing debate about whether MD and H/DX should correlate since the time scales are so dramatically different. When they don't correlate, this difference is invoked as an explanation but in this case they correlate remarkably well. So the local dynamics revealed by MD seem to report on the lower frequency dynamics that the H/DX reports on, in this case. Most important to the conclusions of the paper, the ligand-induced changes in dynamics are transmitted to sites remote from the binding site.

We agree that the timescales can make the direct comparison between MD and HDX data challenging. However, qualitatively these two independent analyses of the dynamics of AcrA provide confidence that the ligand-induced changes in dynamics are transmitted to remote sites away from the binding site.

For the experiments described in Figure 4, with the engineered dimer AcrASD, the recovered values for KD, kon, and koff should be reported. Was the KD determined from the kinetic parameters or from the equilibrium RU plateaus shown in the Figure, or both? In fact the authors are using SPR in a 'nontraditional way to get stoichiometry information about binding which comes from the equilibrium RU values. What is the actual calculated stoichiometry based on this?

We do not report stoichiometry and kinetic information, because the binding reactions are significantly below saturation and the compound is not soluble in water above 200 μ M final concentration used in experiments. The goal of the SPR experiment was to determine whether AcrASD binds the inhibitor and we have modified the main text to reflect this. The analysis of stoichiometry and careful kinetic measurements will require a different set-up. As this qualitative assessment forms more of a supporting characterisation for the AcrASD construct then we have moved the SPR traces to Figure S11, alongside the thermal stability evaluation by circular dichroism.

For the comparison of the H/DX to the MD simulations, the authors refer in the figure legend 4 to 'red' regions of decreased protection upon binding - however, there are no red regions, correct? Should the reference to red regions be left out?

We have modified the caption to indicate the range is from -2 Å (blue) to 0 Å (white).

Also, the authors invoke SAS and show some red regions of increase SAS, but this is not consistent with the H/DX in that region. Do the authors have an explanation?

This is an excellent point and something we had not addressed in the manuscript. For HDX to occur then a free and solvent accessible backbone amide hydrogen is required; to be free to exchange then there must be breakage of the intra- and/or inter-molecular hydrogen bonding network on the amide. Therefore, the transient breakage of these bonds caused by structural dynamics and large domain movements is inextricably linked to solvent accessibility. Upon closer examination of the

trajectories, we notice that the MP domain occasionally comes closer to the $\alpha\beta$ barrel domain in the ligand-bound simulations compared to the apo simulations. This has the consequence of exposing a few residues in the MP domain to solvent more in the bound simulations, due to the greater bend in part of the structure. In areas where we observed increased SAS but reduced RMSF and HDX then this supports that the increased restriction of the protein backbone in this region is likely dominant, even though increased solvent accessibility is detected. The MD simulations help us to disentangle this, and we have included the below sentences in the main text (page 9) to provide further explanation of these results:

‘Interestingly, some areas of the MP, lipoyl and $\alpha\beta$ domains experienced increased SASA in the MD simulations but showed reduced RMSF or HDX (Fig. S5). This observation supports that, even with increased solvent accessibility, enhanced protection of the backbone amide to HDX in these regions is likely dominated by its restricted structural dynamics.’

However, to simplify the presentation, we have moved the figure showing solvent-accessible surface area to the Supporting Information.

For Figure 5, concerning the functional transport properties of the covalently adducted mutants, the axes labels are confusing. It looks like the reported parameter is rate of Hoescht accumulation rather than rate of efflux, so its inversely related to AcrA function. That is OK, but it took a little effort to figure out what was plotted. So, please explicitly report the rates as micromolar accumulated/sec. Also, related to this, it isn't clear how the values on the y-axis are obtained. The authors refer to fitting the fluorescence or absorbance of Hoescht to a double exponential. Why do the kinetics behave as a double exponential, and which of the two rate constants is used in the plots in Figure 5, and why? How does the other rate constant behave? In short the it seems like a lot has been done to the raw data to get the plots and more explanation of this analysis is required.

The Y-axis labels were changed to “Initial rates of HT accumulation ($\mu\text{M/s}$)”. The details of this assay and modeling are described in detail in Ref. 49 by Westfall et al. We included time courses for the accumulation of HT in *E. coli* $\Delta 9$ -Pore cells carrying the plasmid-borne WT AcrAB and the empty pUC18 vector, as examples. We also provided a brief description of data fitting in the Methods.

In general, the results described with the cell-based efflux, or lack of accumulation, don't really report on any dynamic properties of AcrA. The authors are trying to make a correlation between drug-dependent loss of flexibility and loss of function but these assays don't probe dynamics. I believe that covalent adduction in the linker regions could prevent conformational changes required for function but the data don't reveal any change in dynamics, as the H/DX results do with the NSC60339. Interestingly the adducts must be fluorescent and fluorescence polarization might be exploited as a direct readout of dynamics, although I admit I'm not sure what the control sample would be. Alternatively, why not do H/DX on the 6RG-adducted proteins to see how the dynamics are affected. At the very least, the discussion should be rewritten to minimize the connection between the presumed affect of adduction on dynamics and function and just conclude that intact structure in the two linker regions is critical for function.

The reviewer makes a good point about the cell-based efflux assay. We agree that the results of this assay do not directly report on dynamics, and our language used in this section needs to be altered. Whilst the suggestion of further experiments HDX to elucidate dynamics information is exciting, this would be outside the scope of the current study. The discussion (and a line in the introduction) has

been amended to minimize the connection between the presumed effect of adduction on dynamics, and more accurately reflect what the data tells us:

'To determine whether targeting the flexible hinge regions of AcrA could lead to inhibition of antibiotic efflux by AcrAB-TolC *in vivo*, we next developed a covalent efflux inhibition assay using live *E. coli* cells. For this purpose, we constructed the plasmid borne *acrAB* with *acrA* variants containing unique Cys substitutions in the flexible linkers between the MP and $\alpha\beta$ barrel (Leu50Cys, Ile52Cys, Arg225Cys, Glu229Cys, Asn232Cys) and between the $\alpha\beta$ barrel and lipoyl domain (Arg183Cys, Thr205Cys, Asp284Cys) (Fig. 5C). Cells producing the WT and AcrA(Cys)B variants were treated with a Cys-reactive probe MTS-rhodamine 6G (MTS-R6G). This probe is a substrate of AcrAB-TolC and its covalent binding to Cys residues of AcrA located in positions important for conformational flexibility and/or function of the protein is expected to inhibit efflux activity of AcrAB-TolC pump. In this assay, we used hyperporinated *E. coli* $\Delta 9$ -Pore cells lacking all nine TolC-dependent transporters and carrying the plasmid borne AcrA(Cys)AcrB variants.⁴⁶ Hyperporination of the OM has eliminated the permeability barrier of the OM and facilitated MTS penetration into the periplasm to reach AcrA(Cys). Cells were pre-treated with MTS, the unreacted probe washed away and the kinetics of efflux of a fluorescent substrate Hoechst 33342 (Hoechst) was analyzed....

The pre-treatment with MTS-R6G inhibited the activity of AcrA Leu50Cys, Thr205Cys and Asn232Cys variants only, whereas no significant inhibition was seen for other analyzed AcrA(Cys) variants (Fig. 5B and Fig. S15). Hence, only these Cys residues of AcrA are vulnerable to MTS-R6G binding and inhibition, whereas other Cys residues are either inaccessible in the AcrAB-TolC complex or binding of MTS in these sites does not affect the function of the complex. None of these residues are implicated in the characterised AcrA:AcrB binding sites, so the inhibitory effect on efflux is not due to perturbed interactions in the assembly of AcrAB-TolC.⁴⁷ One possibility for this inhibitory effect is that MTS binding causes the reduction in AcrA conformational flexibility, but definite conclusions on restricted dynamics can't be drawn here. However, the results show that these three sites within the flexible hinges between AcrA domains are critical for efflux of the AcrAB-TolC pump.

T205 is in site IV, the proposed binding site of NSC 60339, and further evidences this area is a druggable site for AcrA inhibition. Interestingly, two other residues were mutated in site IV, yet had no effect on efflux (R183 and D284). These residues reside on the periphery of site IV (Fig. 5B), and are likely less critical for AcrA function than T205, which is in the core of the cleft. Furthermore, T205 showed significant protection in the presence of NSC 60339 in both AcrA constructs, highlighting its importance for AcrA flexibility. The two other residues that had an inhibitory effect on efflux are located away from site IV, at the flexible linker between the $\alpha\beta$ -MP domains. N232C and L50C are both located at the bottom of this region and orientated in the same direction (Fig. 5A). The other residues that had no effect on efflux were positioned above N232 and L50 and orientated in the opposite direction. Interestingly, both positions were also found in significantly protected peptides in the presence of NSC 60339 across our HDX-MS experiments. This finding suggests NSC 60339 is stabilising this area downstream from its binding site. Therefore, this provides a novel druggable site on AcrA for the future design of EPs.

We also performed *in vivo* efflux assays with AcrA mutants, to investigate whether deliberately targeting AcrA at the lipoyl- $\alpha\beta$ and the $\alpha\beta$ -MP flexible linkers could affect efflux. This confirmed the lipoyl- $\alpha\beta$ cleft as a druggable site for targeting AcrA during efflux and revealed a possible site between the $\alpha\beta$ -MP domains for future drug design.

Minor points Throughout the manuscript, the word 'data' is used with singular grammar but 'data' is plural. Data 'are' analyzed and 'they' report on events. This should be corrected.

This has been corrected throughout the manuscript.

It would be useful to have the chemical structure of NSC60339 shown somewhere, in addition to the model of it in the docked complex with AcrA.

This has been added to Figure 3 as part C.

Reviewer #3 (Remarks to the Author):

In this study, the authors investigated the molecular mechanism of an inhibitor, NSC 60339, against AcrA using a combination of hydrogen/deuterium exchange mass spectrometry, cellular efflux assays, and molecular dynamics (MD) simulations. Their results demonstrate that the inhibitor could restrict the flexibility of AcrA, by forming a molecular wedge between the lipoyl and $\alpha\beta$ domains. I have some questions and suggestions, focusing on the MD simulations.

1. More details of the Method about molecular dynamics simulations should be provided. The system size of $\sim 476000/906000$ atoms seems to be much larger than that in references 14 and 25. What are the compositions, and why is the large system size required? A detailed figure of the simulation systems should be given in the main text or in the supporting information.

The system was made cubic here in order to allow for tumbling of the protein in all three dimensions. Because of its elongated shape, this required a system significantly larger along two dimensions than would be needed for the static protein. While in previous simulations, we could use an orientational restraint to keep the protein aligned with one axis, NAMD3 does not support this feature yet. Nonetheless, even with a much larger system, NAMD3 is faster than NAMD2.14 on a single GPU. Thus we made the decision to use the larger system without restraints in order to take advantage of NAMD3's performance advantage when possible. This is now noted in the Methods.

We have added Fig. S16 showing AcrA^S, water (156674 molecules), and ions (442 Na⁺ and 443 Cl⁻).

How were the force field parameters of NSC 60339 generated?

Force-field parameters were generated using the CGenFF webserver, now noted in the main text. The topology and parameters are provided in the Supplementary Data.

Did the MD simulations performed at the same pH as the experiments? What are the protonation states of the titratable residues? The protonation states of titratable residues regulate the conformational changes of AcrA (ref 25).

In preparing the system, we checked the pKas of all protonatable residues using propKa. The only residue with a pKa near the experimental pH (6.0) was His285, which was estimated to have a pKa of 5.89, suggesting that even at pH 6.0, it's neutral the majority of the time. We did try simulating once

with His285 protonated, but the ligand rapidly dissociated, likely due to its proximity to the protonated histidine (we note that propKa does not account for bound ligands).

How were the RMSFs calculated? What is the reference structure?

RMSFs were calculated using VMD's "measure rmsf" function. This function uses the average position over the sampled frames as a reference. This is now noted in the Methods.

2.The convergence of the MD simulations should be provided.

The initial structures are from molecular docking or molecular modeling, and may need time to relax. What is more, the main results are the differences of RMSFs between the bound and apo states, the convergence of the MD simulations is essential for comparing different simulations. Therefore, RMSDs and RMSFs of each simulation should be shown in the supporting information at least. And it seems not a good idea to calculate the RMSFs based on the 0-100 ns trajectories.

We have provided the RMSD plots in Figs. S17 (monomer) and S18 (linked dimer). Because of the inter-domain flexibility, we realized that RMSD of the entire protein is not particularly illustrative, so instead, we provide RMSD for each of the four domains separately. We also provide individual RMSF plots in Fig. S6. While in a few cases, a small conformational change within a domain causes the RMSD to rise noticeably compared to others (and then sometimes come back down), the majority appear to stabilize in a reasonable amount of time. Taking the reviewer's advice, we recalculated the RMSF values based on 30-100 ns rather than 0-100 ns. We also now include a fourth replica in order to improve the statistics.

3.Figure 3 is hard to read. The residues, F81 and F254, should be highlighted and labeled in panel B, as well as residues 308-324 in panel C. A color bar should be provided.

The difference is hard to tell from the colors, maybe the values should be given.

We have modified Fig. 3. For the sake of simplifying the presentation, we have removed the original panel D. We have modified panel C (now panel D) to show both F81 and F254 as well as highlight residues 308-324. We also added a colour bar, which we hope makes the differences between regions clearer. We tried modifying panel B to show F81 and F254 but because of the surface representation, they are obscured by other residues.

4.What about the comparison between the results of MD simulations and experiments? The effect from MS results (Figs. 2C and 4C) on AcrA structural dynamics looks different from that from MD simulations (Figs. 3C and 4D).

We agree that there are some noticeable differences between the change in RMSF of AcrA from simulations and the protection from HDX, which no doubt reflects the nine order of magnitude difference in time. This is most apparent in the lipoyl domain which shows significant stabilization in HDX but not in MD. Nonetheless, we believe there are general trends that justify the comparison, namely the broad increase in stability/protection across the entire AcrA induced by binding of the substrate at Site IV near the middle of the protein.

5.Page 8. "However, in MD simulations, both show reduced fluctuations (Fig. 3C) as well as reduced solvent-accessible surface area (SASA; Fig. 3D)," what is the "both" refer to?

Both was referring to F81 and F254. With the addition of a fourth replica and dropping the first 30 ns from analysis, F81 now shows relatively little change in RMSF, so we have modified the sentence to convey that F254 shows reduced fluctuations while both F81 and F254 show a reduction in solvent-accessible surface area.

6. Page 10. "It has also been shown that AcrA lacking the MP domain is still bound by NSC 60339, advocating that its stabilisation is allosteric to NSC 60339 binding to site IV." Is a reference missed in this sentence?

We thank the reviewer for highlighting this - these two necessary references have been added to this sentence.

REVIEWERS' COMMENTS

Reviewer #2 (Remarks to the Author):

The authors have reasonably addressed my concerns and answered my questions. I recommend publication.

Reviewer #3 (Remarks to the Author):

I think the authors has addressed my comments adequately, and I have no more comments.